# SETS: Leveraging Self-Verification and Self-Correction for Improved Test-Time Scaling

**Jiefeng Chen**                                                    *jiefengc@google.com*
*Google Cloud AI Research*

**Jie Ren**                                                              *jjren@google.com*
*Google DeepMind*

**Xinyun Chen**                                                  *xinyunchen@google.com*
*Google DeepMind*

**Chengrun Yang**                                              *chengrun@google.com*
*Google DeepMind*

**Ruoxi Sun**                                                        *ruoxis@google.com*
*Google DeepMind*

**Jinsung Yoon**                                              *jinsungyoon@google.com*
*Google Cloud AI Research*

**Sercan Ö. Arık**                                                *soarik@google.com*
*Google*

**Reviewed on OpenReview:** *https://openreview.net/forum?id=Wv9NMJoKww*

## Abstract

Recent advancements in Large Language Models (LLMs) have created new opportunities to enhance performance on complex reasoning tasks by leveraging test-time computation. However, existing scaling methods have key limitations: parallel methods like repeated sampling are often inefficient and quickly saturate, while sequential methods like SELF-REFINE struggle to improve after a few rounds. Although combining these approaches shows promise, current methods require fine-tuned reward and revision models. This paper proposes *S*elf-*E*nhanced *T*est-Time *S*caling (SETS), a simple yet effective approach that overcomes these limitations by strategically combining parallel and sequential techniques and fully leveraging LLMs' self-improvement abilities. SETS exploits the inherent self-verification and self-correction capabilities of LLMs, unifying sampling, verification, and correction within a single framework. This facilitates efficient and scalable test-time computation for enhanced performance on complex tasks without any model training. Our comprehensive experimental results on challenging benchmarks spanning planning, reasoning, math, and coding demonstrate that SETS achieves significant performance improvements and more advantageous test-time scaling behavior than the alternatives.

## 1 Introduction

Large Language Models (LLMs) have revolutionized artificial intelligence by demonstrating remarkable capabilities in planning, reasoning, coding, and problem-solving across diverse tasks (Team et al., 2024; Touvron et al., 2023; Achiam et al., 2023; anthropic, 2024). Their success stems not only from "training scaling", i.e., their ability to leverage vast datasets and computational resources during training (Kaplan et al., 2020), but also from their ability to benefit from increased compute at test-time to better address

more challenging queries – commonly referred to as "test-time (inference) scaling" (Snell et al., 2024; Wu et al., 2024).

Conventional test-time scaling approaches fall into two categories: parallel and sequential scaling. The parallel scaling approaches such as repeated sampling (Brown et al., 2024), involve generating multiple candidate solutions and selecting the optimal one using techniques like majority voting or task-specific reward models. While these parallel scaling approaches can be effective in certain scenarios, they have notable limitations. The performance improvements from repeated sampling often quickly plateau as the amount of compute increases (Brown et al., 2024). Also, the reliance on task-specific reward models (Christiano et al., 2017; Snell et al., 2024) adds significant training overhead, limiting both efficiency and scalability. The sequential scaling approaches such as SELF-REFINE (Madaan et al., 2024) iteratively revise the current response based on the feedback until the response is verified as correct. As we improve the self-verification and self-correction capabilities of LLMs, sequential scaling approaches become more effective. However, sequential scaling cannot effectively scale up test-time compute to further improve the performance since the performance typically saturates quickly as we increase the self-refinement iterations. Sequential scaling methods like SELF-REFINE stop refining an answer once it is verified as correct, which limits them from scaling to an arbitrarily high compute budget.

To effectively enable more optimal scaling for test-time compute with a canonical framework, we propose an alternative approach that strategically combines the parallel and sequential scaling techniques without training any additional models. Such strategies had been under-explored, likely due to the limited effectiveness of self-correction in earlier generations of LLMs (Huang et al.). However, recent advancements in LLMs have led to significantly improved self-verification and self-correction abilities (Team et al., 2024; gem). These improvements present an opportunity to rethink test-time scaling by moving beyond applying parallel and sequential scaling independently, potentially achieving greater efficiency and generalizability in solving complex tasks.

In this paper, we propose *Self-Enhanced Test-Time Scaling* (SETS) that combines both the parallel and sequential scaling with Sampling, Self-Verify and Self-Correct operations to scale test-time compute. We show that this approach yields more effective test-time compute scaling (i.e., achieving higher accuracy with less compute) compared to notable alternatives such as repeated sampling and SELF-REFINE, as demonstrated with recently-developed advanced LLMs. We evaluate SETS on five challenging benchmarks: NATURAL PLAN (Zheng et al., 2024), LiveBench Reasoning (White et al., 2024), MATH 500 (Hendrycks et al., 2021), AIME 2024-2025 (aim), and LiveCodeBench TestOutputPred (Jain et al., 2024). In our experiments, SETS offers a clear advantage in test-time scaling: it maintains higher effectiveness and experiences less fall-off in performance gains, ultimately outperforming alternatives.

In summary, our contributions are as follows:

- We propose SETS, a simple yet effective method that improves the efficiency of test-time compute scaling for LLMs by leveraging the inherent self-verification and self-correction capabilities of LLMs and combining parallel and sequential scaling techniques.

- We perform extensive experiments to demonstrate that SETS outperforms parallel scaling methods like repeated sampling and sequential scaling approaches like SELF-REFINE, achieving up to 10.9% accuracy improvement on the planning, reasoning, math and coding benchmarks with both non-thinking and thinking models. These results highlight SETS's effectiveness for complex reasoning tasks.

- We conduct ablation studies to analyze the impact of key hyperparameters, such as the maximum number of self-correction rounds and the temperature used during LLM inference, on the performance of SETS. The results indicate that SETS is robust to these settings and achieves strong performance with minimal hyperparameter tuning.

## 2 Related Work

**Test-Time Scaling.** Recent studies have explored leveraging additional test-time compute to enhance the performance of LLMs (Welleck et al., 2024). There are mainly two kinds of test-time scaling approaches:

parallel and sequential scaling (Balachandran et al., 2025). Parallel scaling samples multiple responses from the same model and then aggregates them to obtain a final result through different operators such as majority voting or reward model scoring (Brown et al., 2024). Sequential scaling iteratively improves the response utilizing the feedback of the same model until the response is verified as correct (Madaan et al., 2024). When process-based verifier reward models are available, we can also scale test-time compute by searching against the reward models (e.g., Beam Search and Look-ahead Search (Snell et al., 2024)). We study test-time scaling without utilizing external reward models. We propose a simple yet effective method that combines both parallel and sequential scaling to achieve better test-time scaling performance than those conventional approaches that apply parallel or sequential scaling alone. While Snell et al. (2024) also explored combining parallel sampling and sequential revisions to improve test-time scaling, their approach was limited by the need to train task-specific verifiers and revision models. This dependency may not be practical in real-world scenarios due to the high cost of collecting additional training data. Furthermore, our evaluation is more comprehensive. Unlike Snell et al. (2024), which only tested their method on the MATH benchmark with a single model (PaLM 2-S), our proposed method, SETS, is evaluated on six diverse and challenging benchmarks spanning planning, reasoning, math, and coding. We also test with both "non-thinking" and "thinking" models, which more thoroughly demonstrates the generalization and robustness of our approach. For a more detailed comparison, please see Appendix F.

**Self-Verification.** Verification or reward models play a crucial role in scaling inference compute. Traditional approaches often involve training additional verifiers (Cobbe et al., 2021; Li et al., 2022; Lightman et al., 2023; Liang et al., 2024). More recently, studies showed that LLMs possess the ability to self-verify their outputs (Weng et al., 2023; Song et al., 2024; Zhao et al., 2025). Our work builds on this insight, demonstrating that scaling test-time compute can be significantly enhanced by leveraging LLMs' self-verification performance, particularly for complex reasoning tasks.

**Self-Correction.** Recent research showed that LLMs can refine their solutions to improve performance using either external feedback (Gou et al.), self-feedback (Madaan et al., 2024; Cook et al., 2024; Ferraz et al., 2024), or oracle evaluation (Lee et al., 2025). However, Huang et al. observed that LLMs often struggle to self-correct their responses without external feedback. Qu et al. (2024) proposed an iterative fine-tuning procedure to teach the model to refine its response by recursively detecting and correcting its previous mistakes where the model was trained on a collection of multi-turn data on the domain of math. Our work shows that self-correction, guided by self-verification, can effectively scale test-time compute and significantly improve performance on complex reasoning tasks for advanced LLMs.

**Test-Time Scaling Laws and Model Sizes.** The trade-off between model sizes and test-time compute allocation is of paramount interest. Wu et al. (2024) examined the trade-off between model sizes and generating additional tokens using strategies such as greedy search, majority voting, and Best-of-N. It demonstrated that a small model with advanced inference algorithms can outperform larger models given the same computation budget. Zhang et al. (2024) extended the study from scaling a single LLM to a mixture of multiple LLMs, and proposed an algorithm to find the optimal compute allocation among the mixture, customized for a given task. Chen et al. (2024) observed that in multiple-choice QA tasks, the scaling law based on majority vote only holds for easy queries but not for hard queries. We also study how the scaling law behaves differently for different models, as well as at different difficulty levels of the queries, when self-verification and self-correction are utilized at test-time.

## 3 Method

We introduce *Self-Enhanced Test-Time Scaling* (SETS) framework, which aims to improve accuracy of LLM-generated responses by strategically applying more compute at test time. We leverage the inherent self-verification and self-correction capabilities of LLMs and combine parallel and sequential scaling techniques to achieve better test-time scaling performance. We consider three core operations in the design: Sampling, Self-Verify, and Self-Correct, as shown in Figure 1.

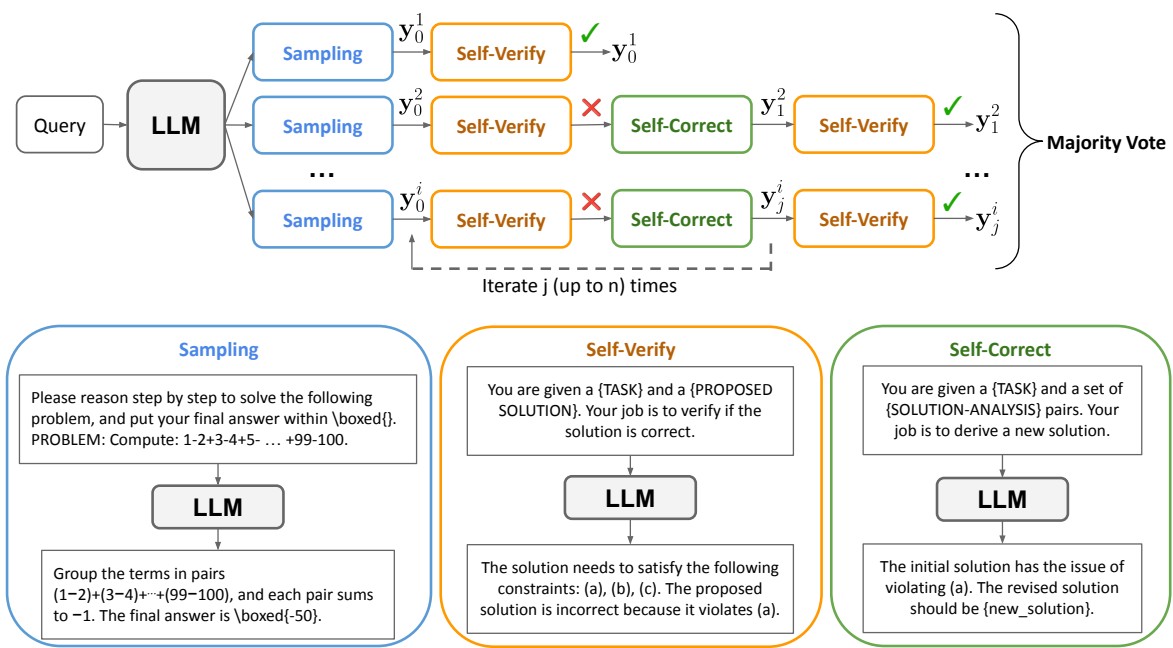

Figure 1: Illustration of the Self-Enhanced Test-Time Scaling (SETS) framework. SETS integrates the Sampling, Self-Verify, and Self-Correct operations to efficiently scale test-time computation.

Each operation is associated with its own prompt. We denote the prompt for Sampling as $I_s(\mathbf{x})$, the prompt for Self-Verify as $I_v(\mathbf{x}, \mathbf{y})$, and the prompt for Self-Correct as $I_c(\mathbf{x}, \{\mathbf{y}_k, \mathbf{r}_k\}_{k=0}^{j})$, where $\mathbf{x}$ is a query, $\mathbf{y}_k$ is a proposed solution for $\mathbf{x}$, and $\mathbf{r}_k$ represent the feedback obtained from the self-verification process for $\mathbf{x}$ and $\mathbf{y}_k$. Suppose $\mathcal{F}$ is an LLM that takes a prompt as input and outputs a response. Then, we have $\mathbf{y} \sim \mathcal{F}(I_s(\mathbf{x}))$, $\mathbf{r} \sim \mathcal{F}(I_v(\mathbf{x}, \mathbf{y}))$ and $\mathbf{y}_{j+1} \sim \mathcal{F}(I_c(\mathbf{x}, \{\mathbf{y}_k, \mathbf{r}_k\}_{k=0}^{j}))$. The feedback $\mathbf{r}$ indicates whether the solution $\mathbf{y}$ is correct or not. We define a judgement function $J(\mathbf{r})$:

$$J(\mathbf{r}) = \begin{cases} 1 & \text{If } \mathbf{y} \text{ is self-verified as correct} \\ 0 & \text{Otherwise.} \end{cases} \tag{1}$$

We adopt the rule-based approach to determine the value of $J(\mathbf{r})$, e.g., if $\mathbf{r}$ contains the string "solution is incorrect", then $J(\mathbf{r}) = 0$; otherwise, $J(\mathbf{r}) = 1$.

SETS judiciously combines Sampling, Self-Verify, and Self-Correct operations to yield superior scaling of test-time computation, as overviewed in Figure 1 and in Algorithm 1. SETS first samples $m$ responses through repeated sampling as the initial set of responses, denoted as $\mathbf{y}_0^1, \mathbf{y}_0^2, \ldots \mathbf{y}_0^m$. For the $i$-th initial response $\mathbf{y}_0^i$, SETS iteratively applies the Self-Verify and Self-Correct processes up to $n$ times to improve the response until it is self-verified as correct, resulting in the improved response $\mathbf{y}^i$. If it reaches the maximum number of self-correction rounds and the response is still self-verified as incorrect, we use the response after $n$ rounds self-correction as $\mathbf{y}^i$. After applying the Self-Verify and Self-Correct process for each of the initial responses, a new set of responses are obtained as $\mathbf{y}^1, \ldots, \mathbf{y}^m$. Majority voting is then used to select the final solution $\mathbf{y}^*$. Suppose we have an indicator function $\mathbb{I}(\mathbf{y} = \mathbf{y}')$ to determine whether two responses $\mathbf{y}$ and $\mathbf{y}'$ are equivalent or not, then:

$$\mathbf{y}^* = \operatorname*{arg\,max}_{\mathbf{y} \in \{\mathbf{y}^1, \ldots, \mathbf{y}^m\}} \frac{1}{m} \sum_{i=1}^{m} \mathbb{I}(\mathbf{y}^i = \mathbf{y}), \tag{2}$$

where we break the tie randomly. The indicator function can be simple exact matching or using LLM-as-a-Judge to determine the equivalence of two responses. In this work, we use the simple exact matching since the benchmarks have a well-structured answer format.

---

**Algorithm 1** SETS: Self-Enhanced Test-Time Scaling

---

**Input:** The query $\mathbf{x}$, the LLM $\mathcal{F}$, the Sampling prompt $I_s$, the Self-Verify prompt $I_v$, the Self-Correct prompt $I_c$, the number of samples $m$, the maximum number of rounds $n$, the judgement function $J$ and the indicator function $\mathbb{I}$.

1: **for** $i = 1, \ldots, m$ **do**
2:     $y_0^i \sim \mathcal{F}(I_s(\mathbf{x}))$                                             {Sampling Operation}
3:     **for** $j = 0, \ldots, n - 1$ **do**
4:         $\mathbf{r}_j^i \sim \mathcal{F}(I_v(\mathbf{x}, \mathbf{y}_j^i))$                                      {Self-Verify Operation}
5:         **if** $J(\mathbf{r}_j^i) = 1$ **then**
6:             $\mathbf{y}^i = \mathbf{y}_j^i$
7:             **Break**                              {Self-Verified as Correct → Early Stop}
8:         **else**
9:             $\mathbf{y}_{j+1}^i \sim \mathcal{F}(I_c(\mathbf{x}, \{\mathbf{y}_k^i, \mathbf{r}_k^i\}_{k=0}^j))$              {Self-Correct Operation}
10:         **end if**
11:         **if** $j = n - 1$ **then**
12:             $\mathbf{y}^i = \mathbf{y}_n^i$
13:         **end if**
14:     **end for**
15: **end for**
16: $\mathbf{y}^* = \arg\max_{\mathbf{y} \in \{\mathbf{y}^1, \ldots, \mathbf{y}^m\}} \frac{1}{m} \sum_{i=1}^m \mathbb{I}(\mathbf{y}^i = \mathbf{y})$                      {Majority Voting}
**Output:** The final solution $\mathbf{y}^*$.

---

SETS utilizes the LLM directly, integrating parallel and sequential scaling techniques to enhance the efficiency of test-time compute scaling, especially when ample compute budget is available. The sequential scaling method SELF-REFINE Madaan et al. (2024) can be regarded as a special case of SETS (when $m = 1$). However, SELF-REFINE cannot effectively scale up test-time compute since it terminates when the stopping condition is met. Therefore, while SELF-REFINE is primarily effective in low-compute budget regimes, SETS demonstrates strong performance in high-compute budget regimes as well. Our experiments across a wide range of scenarios confirm this (see Section 4.3).

## 4 Experiment

### 4.1 Scaling Laws for Test-Time Compute

We define test-time compute-optimal scaling as the strategy that selects hyperparameters $\theta$ for a given approach to maximize performance within a compute budget $C$ on a specific dataset $\mathcal{D}$ and LLM $\mathcal{F}$:

$$\theta^*(C|\mathcal{D}, \mathcal{F}) = \arg\max_{\theta \in \Theta} M(\theta|\mathcal{D}, \mathcal{F}), s.t. H(\theta) \leq C, \tag{3}$$

where $\Theta$ are candidate values of hyperparameters for the test-time strategy, $H$ is the cost function that maps hyperparameters $\theta$ to the average amount of compute used for each input (e.g., average number of output tokens), and $M$ is a performance metric such as accuracy. For example, $\theta$ in the proposed method SETS contains two variables $m$ and $n$. We obtain the scaling law curve with the x-axis corresponding to budget $C$ and the y-axis corresponding to performance $M(\theta^*(C|\mathcal{D}, \mathcal{F}))$. To compute each point $(x, y)$ on the scaling curve, we first consider a specific cost $x = H(\theta)$. For this cost, we find the optimal performance $y = M(\theta^*(x|\mathcal{D}, \mathcal{F}))$ evaluating all hyperparameter configurations within $\Theta$. Finally, adjacent points are connected to generate the scaling law curve.

### 4.2 Setup

**Datasets.** We experiment on six datasets that contain complex instructions and require advanced reasoning for accurate responses: Trip Planning and Meeting Planning in NATURAL PLAN (Zheng et al., 2024), LiveBench Reasoning (White et al., 2024), MATH 500 (Hendrycks et al., 2021), AIME 2024-2025 (aim),

and LiveCodeBench TestOutputPred (Jain et al., 2024). The details of these benchmarks can be found in Appendix A. Since the ground truth answers across all tasks are well-structured and can be verified either by exact match or rule-based checker, we do not need any model based evaluator to evaluate the accuracy of the model-generated responses.

**Prompts.** We design tailored prompts for three key operations – Sampling, Self-Verify, and Self-Correct (provided in Appendix B) to enable these operations using LLMs. We use existing templates if available or create simple and direct prompts, to generalize across tasks and models as much as possible. For NATURAL PLAN tasks, we use controlled generation with Langfun (Peng, 2023) to obtain structured solutions to improve accuracy for all methods (refer to Appendix C for details). We do zero-shot prompting for Self-Verify and Self-Correct – using only instructions without including any few-shot examples.

**Baselines.** For fair comparison, we adopt the following baselines that don't need additional model training or external reward models. We use the same prompts for Sampling, Self-Verify, and Self-Correct described in Appendix B for the baselines. BoN stands for Best-of-N (i.e. sample multiple responses and choose one using some mechanisms as the final response).

- **SELF-REFINE**: One single initial solution is sampled and then is iteratively refined via Self-Verify and Self-Correct processes up to $n$ times to improve the response until it is self-verified as correct (Madaan et al., 2024). Note that SELF-REFINE cannot arbitrarily scale up the test-time compute because it could early stop as long as the solution is self-verified correctly. SETS addresses this limitation by integrating parallel sampling, allowing for greater scalability.

- **BoN+Majority Vote**: We sample $m$ solutions and then perform majority voting via exact matching on the sampled solutions to select the most frequent solution (also referred as Self-Consistency (Wang et al., 2022)). No self-verify or self-correction is involved.

- **BoN+Self-Eval**: Similar to BoN+Majority Vote, we sample $m$ solutions and then query the LLM to select the final solution with a multi-choice QA task prompt (described in Appendix B.4) as used in Ren et al. (2023).

- **BoN+Self-Verify**: We sample $m$ solutions and self-verify each one, then perform a majority vote via exact matching on the solutions verified as correct to select the final solution. If all sampled solutions are verified as incorrect, we perform a majority vote on all sampled solutions. No self-correction is involved.

To summarize, our proposed method SETS integrates all three components of parallel Sampling, Self-Verify, and Self-Correct, while the baselines are either missing 1 or 2 components, as shown in Table 1.

| Method | Sampling | Self-Verify | Self-Correct |
|---|:---:|:---:|:---:|
| SELF-REFINE | ✗ | ✓ | ✓ |
| BoN+Majority vote | ✓ | ✗ | ✗ |
| BoN+Self-Eval | ✓ | ✗ | ✗ |
| BoN+Self-Verify | ✓ | ✓ | ✗ |
| SETS (ours) | ✓ | ✓ | ✓ |

Table 1: Comparison of different baselines with SETS

**LLMs and Configs.** Our experiments utilize both proprietary and open-source models, which include both "non-thinking" and "thinking" types. The non-thinking models include GEMINI-1.5-Pro-002, Claude-3.5-Sonnet-20241022, Qwen3-235B-A22B, and Qwen2.5-1.5B-Instruct while the thinking models include GEMINI-2.5-Flash-Lite-Thinking and GEMINI-2.5-Flash-Preview-04-17. Qwen3-235B-A22B and Qwen2.5-1.5B-Instruct are open-source models while the others are proprietary models. For GEMINI-2.5-Flash-Lite, we set the thinking budget to $24,576$ to turn on thinking. We use a temperature of $0.7$ to perform three

operations Sampling, Self-Verify and Self-Correct for all models. For BoN+Self-Eval, we use a temperature of 0.7 for sampling multiple responses and then use a temperature of 0 for the final self-evaluation step (i.e., selecting the best answer among the responses).

**Hyperparameter Set ($\Theta$).** To find the maximum performance at a given compute budget, we search across different hyperparameter settings (i.e., the set of candidate hyperparameters $\Theta$). For SELF-REFINE, $\theta \in \Theta$ has one hyperparameter – the number of refinement iterations $n$ and we set $n \in [1, 10]$. We don't consider larger $n$ because the refinement process typically stops before 10 iterations. For BoN approaches, $\theta \in \Theta$ has one hyperparameter – the number of samples $m$. We set a sufficiently large value for $m$ so that further increases do not yield significant accuracy improvements. For baselines BoN (Majority Vote or Self-Eval), we set $m \in [1, 100]$ for non-thinking models while setting $m \in [1, 50]$ for thinking models. For thinking models, the value of $m$ is halved because their output length is generally much longer. For the proposed method SETS, $\theta \in \Theta$ has two hyperparameters – the number of samples $m$ and the maximum number of rounds $n$ of Self-Verify and Self-Correct. We set $m \in [1, 50] \wedge n \in [1, 10]$ for non-thinking models and set $m \in [1, 25] \wedge n \in [1, 10]$ for thinking models to balance between the compute allocated to sampling and self-improvement. For baseline BoN+Self-Verify, we define $m \in [1, 50]$ for non-thinking models and set $m \in [1, 25]$ for thinking models. The maximum value of $m$ for SETS and BoN+Self-Verify is halved compared to BoN+Majority Vote and BoN+Self-Eval to ensure comparable maximum compute budgets across them.

**Compute Cost Estimation.** Since different operations (Sampling, Self-Verify, Self-Correct) use different prompts and generate different lengths of responses, to make fair comparison, we focus on the average number of output tokens to estimate the cost (as the price for output tokens is much higher than that for input tokens[1]). We also provide results based on the number of API calls (Appendix D.1) and financial cost (Appendix D.9). For our cost analysis, we deliberately avoid using wall-clock time, as it is highly volatile and influenced by uncontrollable factors such as network latency, API server load, and hardware specifics. Our chosen metrics provide a standardized, hardware-agnostic basis for comparison that reflects the intrinsic efficiency of each method and ensures the reproducibility of our results.

### 4.3 Results

**Improved Test-time Scaling with SETS.** SETS consistently outperforms the baselines (Figure 2) across different benchmarks, yielding increased accuracy gains as the test-time compute increases for GEMINI-1.5-Pro. For BoN with Majority Vote, the accuracy typically saturates quickly with the increase in the amount of test-time compute. While BoN combined with Self-Verify or Self-Eval yields better results than BoN with Majority Vote on some tasks, it does not show consistent improvement across all tasks. In contrast, SETS utilizes both self-verification and self-correction, yielding accuracy improvements across all datasets. These findings are consistent when using the number of API calls as the measure of compute cost (see Appendix D.1).

**Impact of Different LLMs.** Besides GEMINI-1.5-Pro, we also apply SETS with other LLMs: Gemini-2.5-Flash, Gemini-2.5-Flash-Lite, Claude-3.5-Sonnet, Qwen3-235B-A22B, and Qwen2.5-1.5B-Instruct. Figure 3 shows that for those LLMs, SETS still outperforms the baselines on most of the cases with a few exceptions. We hypothesize that the performance of SETS is affected by the models' self-verification and self-correction capabilities. So we evaluate the accuracy of self-verification and self-correction individually to disentangle their effects. To evaluate the self-verification performance, we ask the LLM to self-verify its own proposed solution (sampled with temperature$= 0$) and evaluate whether we can use the verification result to detect errors (treating the error as the positive class, we calculate the precision, recall, and F1 score). To evaluate the self-correction performance, we ask the LLM to self-correct the proposed solution up to 2 rounds (using the SELF-REFINE algorithm). The results are shown in Table 2. Comparing Figure 2, 3 and Table 2, we observe that when the model has strong self-verification and self-correction performance, SETS can significantly outperform the baselines. However, when the models' self-verification and self-correction performance is weak, SETS might not provide significant gains (e.g., Claude-3.5-Sonnet on LiveBench Reasoning). Appendix D.3 shows that increasing the sample size for self-verification and applying majority voting can improve the

---

[1] https://ai.google.dev/pricing, and https://www.anthropic.com/pricing

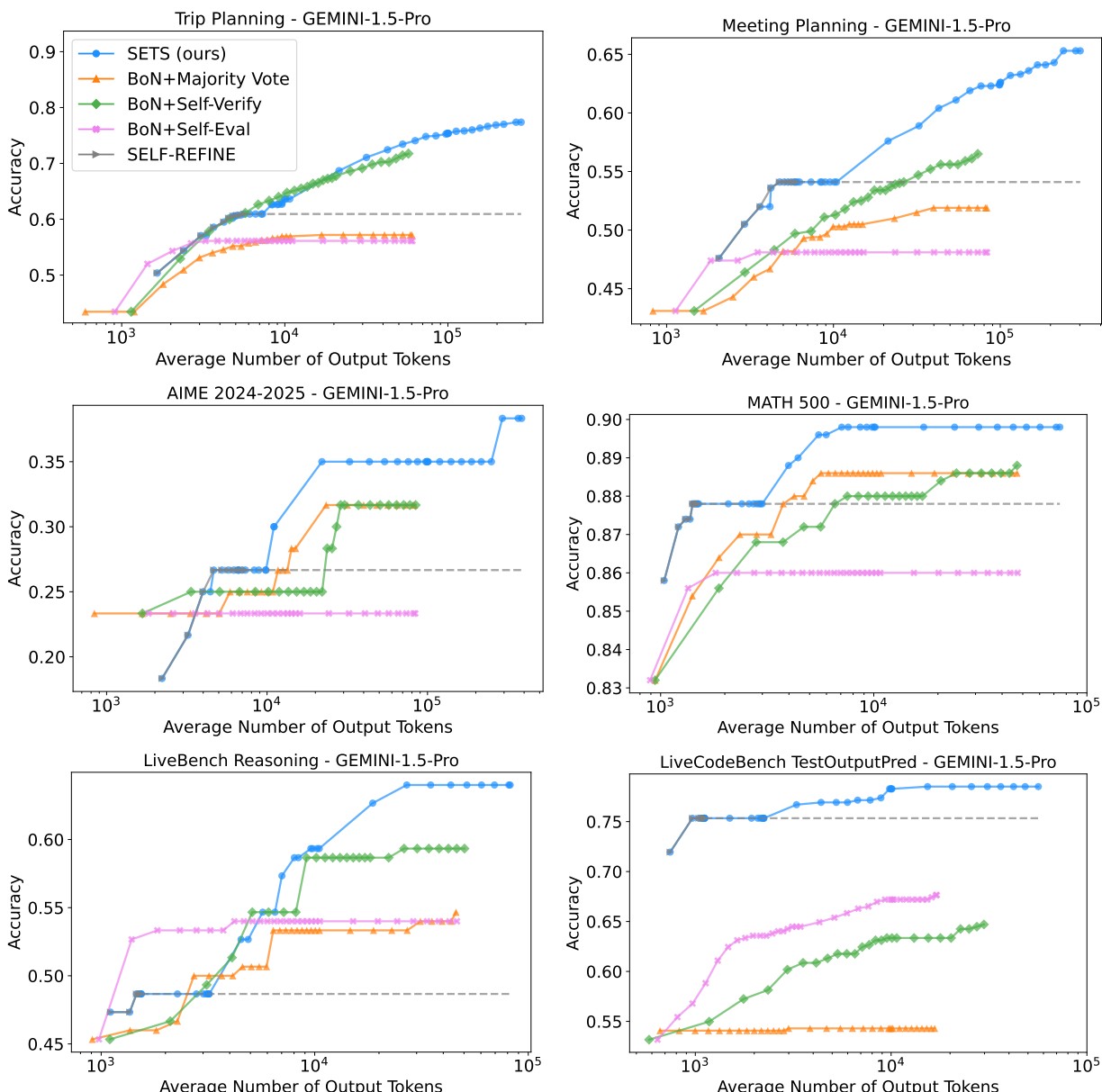

Figure 2: Scaling law curves where the x-axis is the average number of output tokens and y-axis is the accuracy. Each point $(x, y)$ in the curve corresponds to a hyperparameter setting $\theta \in \Theta$. $y$ is the optimal performance at the cost budget $x = H(\theta)$ (see Section 4.1 for details). We subsample the points (up to 8 within every x-tick interval) to make the markers less crowded. SELF-REFINE would early stop if the solution is self-verified correctly, so it can not scale up arbitrarily as shown in dotted line.

self-verification accuracy, which aligns with the findings in Zhao et al. (2025). Appendix D.12 provides a qualitative analysis of self-verification's failure modes.

**The Effect of Self-Correction Rounds.** We study whether allocating more test-time compute to Self-Verify and Self-Correct leads to better end-to-end accuracy given a fixed test-time compute budget. The hyperparameter of the maximum number of rounds ($n$) in SETS controls the compute allocated to Self-Verify and Self-Correct. Given a fixed compute budget, a larger number of rounds $n$ suggests a smaller number of samples $m$. Figure 4 shows that given a fixed compute budget, increasing the number of rounds of Self-Verify

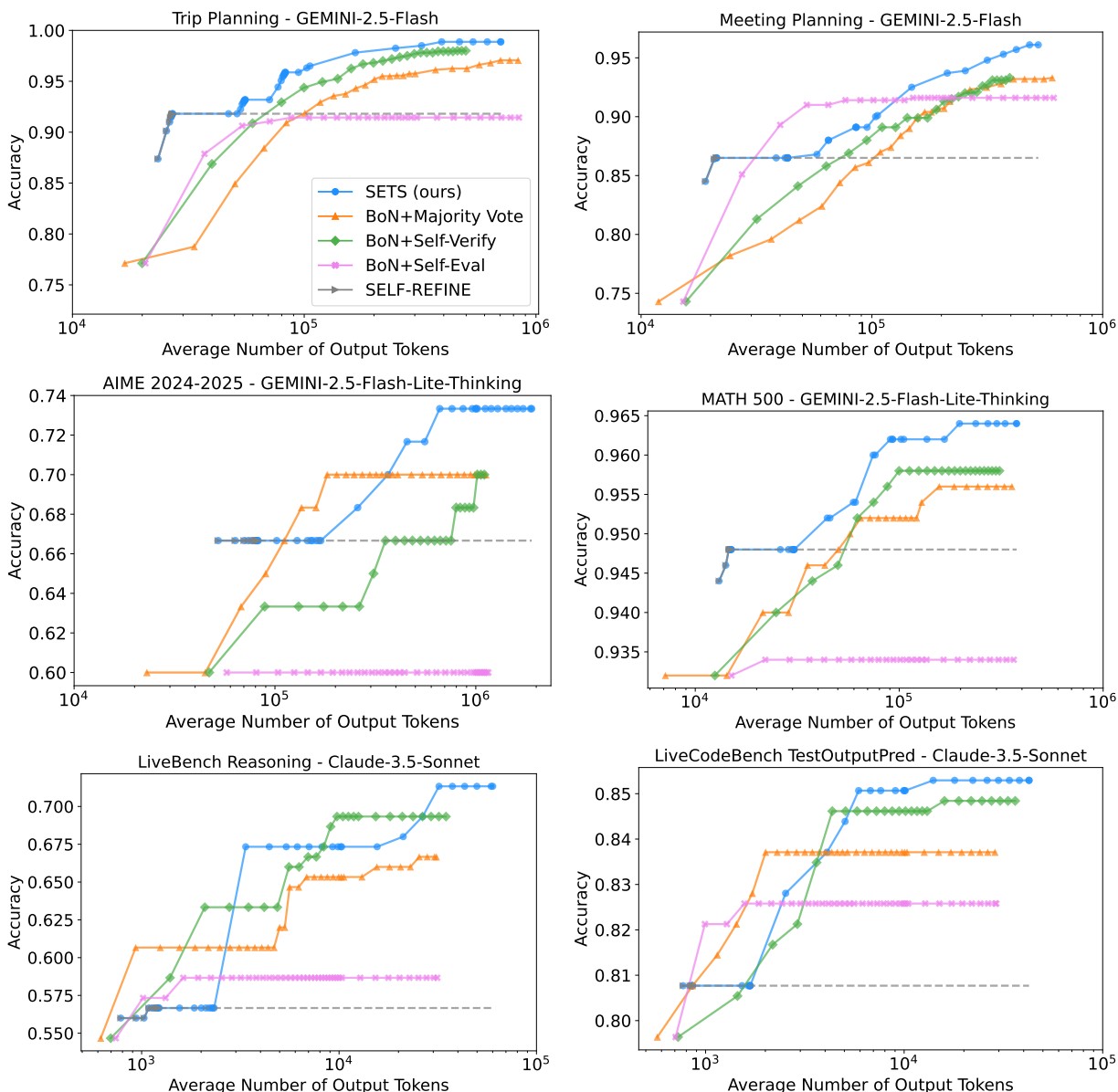

Figure 3: Scaling law curves with various LLMs (Gemini-2.5-Flash, GEMINI-2.5-Flash-Lite-Thinking and Claude-3.5-Sonnet). The complete results for all datasets and LLMs are provided in Appendix D.2.

and Self-Correct generally leads to accuracy gains, although the impact varies across tasks. For Trip Planning and Meeting Planning, the accuracy increases as the number of rounds increases, but the returns diminish after $n = 4$. Based on the results, we can set a sufficiently large value for $m$ (e.g., $m = 50$) and set $n = 4$ for SETS to achieve strong performance in practice.

**The Effect of Temperature for SETS.** We study how the temperature used for the three core operations (Sampling, Self-Verify, and Self-Correct) affects the performance of SETS. We consider two configurations: (1) using a temperature of 0.7 for all three operations (our default setting), and (2) using a temperature of 0.7 for Sampling, but a temperature of 0.0 (greedy decoding) for Self-Verify and Self-Correct. The results in Figure 5 show that our default setting generally achieves better performance across different benchmarks. This suggests that introducing a higher degree of randomness (temperature = 0.7) for the Self-Verify and Self-Correct operations is beneficial. The increased temperature likely promotes a broader exploration of

| Dataset | Model | Self-Verification Evaluation | | | Self-Correction Evaluation | | |
|---|---|---|---|---|---|---|---|
| | | Precision | Recall | F1 Score ↑ | Initial Accuracy | Round 1 Δ ↑ | Round 2 Δ ↑ |
| Trip Planning | Claude-3.5-Sonnet | 93.74 | 91.85 | 92.78 | 41.62 | +4.50 | +8.06 |
| | GEMINI-1.5-Pro | 97.51 | 94.04 | **95.74** | 43.44 | +6.94 | +10.94 |
| | GEMINI-2.5-Flash | 97.15 | 72.09 | 82.77 | 77.12 | **+10.25** | **+13.00** |
| | GEMINI-2.5-Flash-Lite-Thinking | 99.66 | 55.59 | 71.37 | 37.31 | +9.38 | +12.13 |
| Meeting Planning | Claude-3.5-Sonnet | 78.69 | 80.17 | 79.42 | 53.60 | +0.40 | +2.10 |
| | GEMINI-1.5-Pro | 80.30 | 82.29 | 81.28 | 43.10 | +4.50 | +7.40 |
| | GEMINI-2.5-Flash | 74.18 | 66.39 | 70.07 | 74.30 | +10.20 | +12.10 |
| | GEMINI-2.5-Flash-Lite-Thinking | 79.72 | 83.20 | **81.42** | 50.50 | **+10.80** | **+14.90** |
| LiveBench Reasoning | Claude-3.5-Sonnet | 72.22 | 40.00 | 51.49 | 54.67 | +1.33 | +0.00 |
| | GEMINI-1.5-Pro | 77.36 | 53.25 | 63.08 | 45.33 | +2.00 | +1.33 |
| | GEMINI-2.5-Flash | 75.61 | 62.00 | **68.13** | 68.00 | **+11.33** | **+14.67** |
| | GEMINI-2.5-Flash-Lite-Thinking | 90.48 | 52.78 | 66.67 | 69.33 | +2.00 | +8.67 |
| MATH 500 | Claude-3.5-Sonnet | 61.54 | 20.00 | 30.19 | 72.80 | +2.00 | +2.00 |
| | GEMINI-1.5-Pro | 68.12 | 61.84 | **64.83** | 83.20 | **+2.60** | **+4.00** |
| | GEMINI-2.5-Flash | 14.94 | 56.10 | 23.59 | 94.80 | -3.80 | -1.80 |
| | GEMINI-2.5-Flash-Lite-Thinking | 55.29 | 74.60 | 63.51 | 93.20 | +1.20 | +1.40 |
| AIME 2024-2025 | Claude-3.5-Sonnet | 94.74 | 35.29 | 51.43 | 10.00 | +0.00 | +**0.00** |
| | GEMINI-1.5-Pro | 89.19 | 71.74 | 79.52 | 23.33 | -5.00 | -1.67 |
| | GEMINI-2.5-Flash | 29.63 | 38.10 | 33.33 | 73.33 | +1.67 | +0.00 |
| | GEMINI-2.5-Flash-Lite-Thinking | 100.00 | 70.59 | **82.76** | 60.00 | **+6.67** | **+6.67** |
| LiveCodeBench TestOutputPred | Claude-3.5-Sonnet | 76.92 | 30.61 | 43.80 | 79.64 | +1.13 | +1.13 |
| | GEMINI-1.5-Pro | 89.17 | 68.29 | **77.35** | 53.17 | **+18.78** | **+22.17** |
| | GEMINI-2.5-Flash | 16.33 | 45.71 | 24.06 | 95.25 | +0.90 | +1.13 |
| | GEMINI-2.5-Flash-Lite-Thinking | 15.24 | 20.00 | 17.30 | 93.67 | +0.45 | +0.68 |

Table 2: Performance on self-verification and self-correction. Round k Δ means Round k accuracy minus initial accuracy. All numbers are in terms of percentages. **Bold** numbers are superior results.

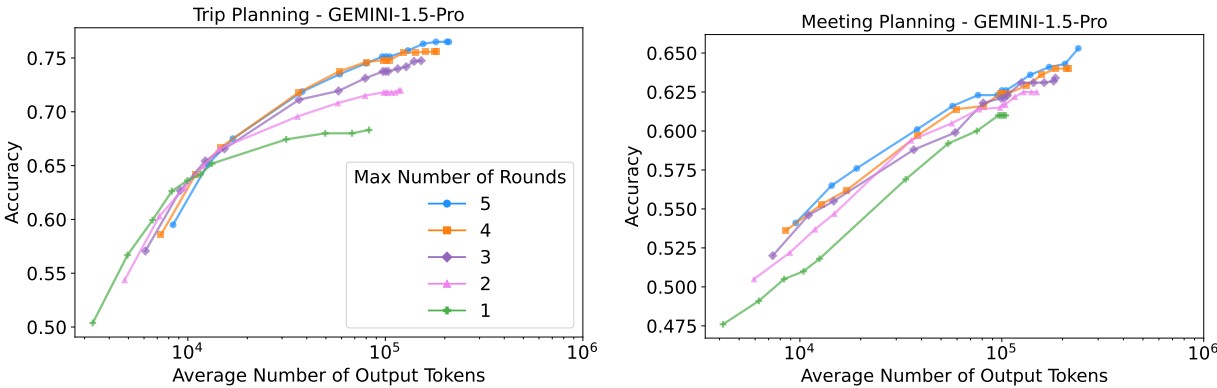

Figure 4: The effect of allocating more compute to self-verification and self-correction for SETS (controlled by max number of rounds) given a fixed computational budget (measured by average number of output tokens). The results for other datasets are provided in Appendix D.4.

alternative reasoning paths, which is crucial for handling complex reasoning tasks. This diversity in thought, combined with the final majority voting mechanism, appears to be a key factor in improving the overall performance and robustness of the SETS framework.

**Non-thinking Mode with SETS vs. Thinking Mode.** SETS functions as a capability amplifier, not a creator of reasoning. We demonstrated this by comparing a "non-thinking" mode with SETS against a superior "thinking" mode with BoN+Majority Vote under a fixed token budget, where the former could not match the latter's performance (Appendix D.10). However, applying SETS to the thinking mode yielded substantial gains, confirming its practical utility is to push a chosen model to its absolute performance limit.

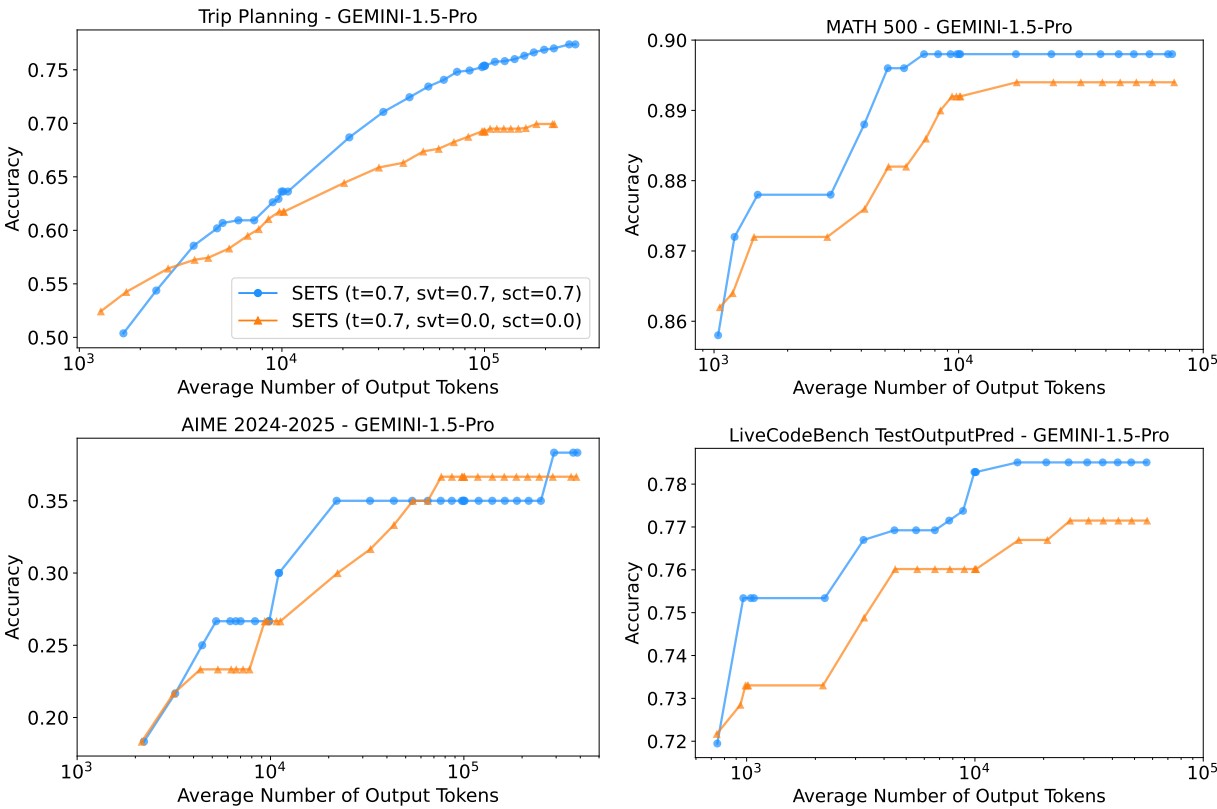

Figure 5: The effect of different temperature settings for SETS. t, svt and sct are temperature parameters for the Sampling, Self-Verify and Self-Correct operations respectively. The results for other datasets are provided in Appendix D.5.

**SETS with Confidence-weighted Voting.** SETS is compatible with diverse aggregation methods, including confidence-weighted voting (Appendix D.11). This strategy generally outperforms standard majority voting, but its effectiveness is task-dependent, with the simpler method sometimes proving superior. This indicates that the reliability of the underlying confidence heuristic can vary across tasks.

## 5 Conclusions

In this paper, we introduced $\boldsymbol{S}$*elf-*$\boldsymbol{E}$*nhanced* $\boldsymbol{T}$*est-Time* $\boldsymbol{S}$*caling* (SETS), a simple yet effective paradigm for scaling test-time compute that capitalizes on the inherent self-verification and self-correction mechanisms of LLMs. SETS uniquely integrates parallel and sequential scaling, distinguishing it from prior work that often relies on specialized fine-tuning. Our experimental results reveal that SETS, by sampling a set of initial responses and then iteratively refining them, surpasses baselines like purely repeated sampling or SELF-REFINE. Importantly, SETS consistently delivers higher quality outputs and demonstrates increasing returns as test-time computation increases across challenging planning, reasoning, math, and coding tasks.

**Limitation.** Our future work will focus on expanding the SETS framework by addressing its current limitations and enhancing its core dependencies. A key priority is to improve the foundational self-critique and self-correction capabilities of LLMs, as the efficacy of SETS is directly tied to these abilities. We anticipate that as LLMs continue to advance, their capacity for self-improvement will likewise strengthen, thus broadening the applicability and effectiveness of SETS. We also aim to enhance the efficiency of SETS for low-resource settings and complement the framework with prompt optimization for models with weaker self-correction skills. While this work concentrates on reasoning tasks with objectively verifiable answers, we plan to extend its applicability to domains like summarization and tool use. This expansion will necessitate a move

from majority voting to more sophisticated aggregation strategies, such as Universal Self-Consistency (Chen et al., 2023). Finally, though our evaluation is currently confined to text-only datasets, the SETS framework is designed for future extension to multi-modal benchmarks.

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

# Appendix

In Appendix A, we provide details about the datasets used in our experiments. Appendix B outlines the prompts designed for Sampling, Self-Verify, Self-Correct, and Multi-choice QA. Appendix C details the controlled generation process. Additional experimental results are presented in Appendix D. Appendix E includes detailed responses for the three core operations employed within SETS. Finally, Appendix F compares the proposed SETS with the Combining Sequential/Parallel approach from Snell et al. (2024).

## A Datasets

We perform experiments on six datasets: Trip Planning and Meeting Planning from the NATURAL PLAN benchmark (Zheng et al., 2024), the LiveBench Reasoning benchmark (White et al., 2024), the MATH 500 benchmark (Hendrycks et al., 2021), AIME 2024-2025 benchmark (aim), and the LiveCodeBench TestOutputPred benchmark (Jain et al., 2024).

NATURAL PLAN provides 5 examples as few-shot exemplars for each task (i.e. the 5-shot setting). NATURAL PLAN also provides a controlled variable (e.g. number of people, number of cities, number of days, etc) that can indicate the difficulty level of each task. We utilize this controlled variable to understand the performance of different methods on easy and hard subset of the NATURAL PLAN datasets. In Trip Planning and Meeting Planning, the ground-truth solutions are long-form and contain multiple steps.

LiveBench Reasoning is a task from LiveBench, which is a benchmark for LLMs designed with test set contamination and objective evaluation in mind. LiveBench Reasoning has three tasks: spatial, zebra_puzzle and web_of_lies_v2, each containing 50 test examples.

MATH 500 is a subset of 500 problems from the MATH benchmark (Hendrycks et al., 2021), which contains 12,500 challenging competition mathematics problems.

AIME 2024-2025 contains problems from the American Invitational Mathematics Examination (AIME) 2024 - 2025. AIME is a prestigious high school mathematics competition known for its challenging mathematical problems.

LiveCodeBench TestOutputPred is a task from LiveCodeBench, which is a holistic and contamination-free evaluation benchmark of LLMs for code. LiveCodeBench focuses on broader code-related capabilities, such as self-repair, code execution, and test output prediction, beyond mere code generation. We use the test output prediction dataset, which contains 442 examples.

We summarize the statistics of these datasets in Table 3.

| Task Type | Benchmark | Number of Test Examples |
|---|---|---|
| Planning | Trip Planning
Meeting Planning | 1600
1000 |
| Reasoning | LiveBench Reasoning | 150 |
| MATH | MATH 500
AIME 2024-2025 | 500
60 |
| Coding | LiveCodeBench TestOutputPred | 442 |

Table 3: The statistics of the datasets used in the experiments.

# B   Prompts

In this section, we present the prompts used for Sampling, Self-Verify, and Self-Correct operations. Our design philosophy for the prompts centered on minimalism and generalizability to demonstrate that our method's performance is robust and not dependent on extensive prompt engineering. We intentionally created simple, standardized templates to show that the core logic of SETS is effective across diverse tasks and models without highly tailored instructions. For example, the Self-Correct prompt uses a direct instruction to "outline your step-by-step thought process for deriving a new solution." For Self-Verify, we found a simple, structured format – asking the model to first "1. List all constraints in the TASK" and then "2. Verify if the PROPOSED SOLUTION satisfies each of the constraints" – was consistently effective at guiding the model's reasoning. This minimalist approach enhances the reproducibility of our method and confirms that its gains stem from its inherent structure rather than from fine-tuned prompts.

## B.1   Sampling Prompt

For NATURAL PLAN benchmarks, we construct the sampling prompt by adding some additional instructions to the original task description prompt.

> **Sampling Prompt for Trip Planning**
>
> {planning_task_description_with_demos}
> Please first list all the constraints in the problem and then output a final solution that satisfies all the constraints.

> **Sampling Prompt for Meeting Planning**
>
> {planning_task_description_with_demos}
> Please first list all the constraints and optimization goals in the problem and then output a final solution that satisfies all the constraints and optimization goals.

For the MATH 500 and AIME 2024-2025 benchmarks, we construct the sampling prompt by adding some additional instructions to elicit the LLM's reasoning and ensure the final answer is boxed.

> **Sampling Prompt for MATH and AIME**
>
> You are an expert in solving math problems. Please reason step by step to solve the following problem, and put your final answer within \boxed{}.
> PROBLEM:
> {problem}

For the LiveBench Reasoning and LiveCodeBench TestOutputPred benchmarks, we use the original prompt provided by the benchmarks as the sampling prompt.

## B.2   Self-Verify Prompt

For the NATURAL PLAN benchmarks, we use the following Self-Verify prompt:

---

**Self-Verify Prompt for NATURAL PLAN**

{planning_task_demos}
You are an expert at {task_type}. You are given a TASK of {task_type} request, and a PROPOSED SOLUTION. Your job is to:
1. List all constraints in the TASK.
2. Verify if the PROPOSED SOLUTION satisfies each of the constraints with justifications.
3. Write a line of the form "The proposed solution is correct" or "The proposed solution is incorrect" at the end of your response based on your analysis.
TASK:
{planning_task_description_without_demos}
PROPOSED SOLUTION:
{solution}

---

For the MATH 500 and AIME 2024-2025 benchmarks, we use the following Self-Verify prompt:

---

**Self-Verify Prompt for MATH and AIME**

You are an expert in solving math problems. You are given a PROBLEM and a PROPOSED ANSWER. Your job is to:
1. Identify all conditions and constraints in the PROBLEM for verifying the correctness of the proposed answer.
2. Reason step by step to verify if the proposed answer satisfies each of the conditions and constraints.
3. Write a line of the form "The proposed answer is correct" or "The proposed answer is incorrect" at the end of your response based on your analysis.
PROBLEM:
{problem}
PROPOSED ANSWER:
{answer}

---

For the LiveBench Reasoning benchmark, we use the following Self-Verify prompt:

---

**Self-Verify Prompt for LiveBench Reasoning**

You are an expert in solving problems that require reasoning. You are given a QUESTION and a PROPOSED ANSWER. Your job is to:
1. Transform the PROPOSED ANSWER into a statement given the QUESTION and identify all constraints in the QUESTION for verifying the statement.
2. Think step by step to verify if the statement satisfies each of the constraints.
3. Write a line of the form "The statement is correct" or "The statement is incorrect" at the end of your response based on your analysis.
QUESTION:
{question}
PROPOSED ANSWER:
{answer}

---

For the LiveCodeBench TestOutputPred benchmark, we use the following Self-Verify prompt:

---

**Self-Verify Prompt for LiveCodeBench TestOutputPred**

You are an expert Python programmer. You will be given a question (problem specification) and a PROSPOSED ANSWER (a Python program). Your job is to:
1. Identify all constraints in the Question for verifying the correctness of the PROPOSED ANSWER.
2. Think step by step to verify if the PROPOSED ANSWER satisfies each of the constraints.
3. Write a line of the form "The proposed answer is correct" or "The proposed answer is incorrect" at the end of your response based on your analysis.
{problem}
PROPOSED ANSWER:
{answer}

---

## B.3 Self-Correct Prompt

For the NATURAL PLAN benchmarks, we use the following Self-Correct prompt:

---

**Self-Correct Prompt for NATURAL PLAN**

{planning_task_demos}
You are an expert at {task_type}. You are given a TASK of {task_type} request. You are also given a set of solution-analysis pairs. Your job is to outline your step-by-step thought process for deriving a new solution.
TASK:
{planning_task_description_without_demos}
{solution_and_analysis}

---

For the MATH 500 and AIME 2024-2025 benchmarks, we use the following Self-Correct prompt:

---

**Self-Correct Prompt for MATH and AIME**

You are an expert in solving math problems. You are given a PROBLEM and a set of answer-analysis pairs. Your job is to reason step by step for getting a correct answer and put your final answer within \boxed{}.
PROBLEM:
{problem}
{answer_and_analysis}

---

For the LiveBench Reasoning benchmark, we use the following Self-Correct prompt:

---

**Self-Correct Prompt for LiveBench Reasoning**

You are an expert in solving problems that require reasoning. You are given a QUESTION and a set of answer-analysis pairs. Your job is to outline your step-by-step thought process for getting a correct answer.
QUESTION:
{question}
{answer_and_analysis}
{answer_trigger}

---

For the LiveCodeBench TestOutputPred benchmark, we use the following Self-Correct prompt:

---

**Self-Correct Prompt for LiveCodeBench TestOutputPred**

You are a helpful programming assistant and an expert Python programmer. You are helping a user to write a test case to help to check the correctness of the function. The user has written a input for the testcase. The user has also provided a set of answer-analysis pairs. Your job is to outline your step-by-step thought process to calculate the output of the testcase and write the whole assertion statement in the markdown code block with the correct output.
{problem}
{answer_and_analysis}

---

## B.4 Multi-choice QA Task Prompt for Self-Evaluation

For the NATURAL PLAN benchmarks, we use the following multi-choice QA task prompt:

---

**Multi-choice QA Task Prompt for NATURAL PLAN**

{planning_task_demos}
You are an expert at {task_type}. You are given a TASK of {task_type} request. You are also given a set of possible solutions. Your job is to outline your step-by-step thought process for selecting the best solution.
TASK:
{planning_task_description_without_demos}
{solution_choices}
The output should be in JSON format: {{"reason": "<your reasoning>", "solution_id": "<an integer between 1 and {num_solution_choices}>"}}

---

For the MATH 500 and AIME 2024-2025 benchmarks, we use the following multi-choice QA task prompt:

---

**Multi-choice QA Task Prompt for MATH and AIME**

You are an expert in solving math problems. You are given a PROBLEM and a set of possible answers. Your job is to reason step by step for selecting the best answer.
PROBLEM:
{problem}
{answer_choices}
The output should be in JSON format: {{"reason": "<your reasoning>", "answer_id": "<an integer between 1 and {num_answer_choices}>"}}

---

For the LiveBench Reasoning benchmark, we use the following multi-choice QA task prompt:

---

**Multi-choice QA Task Prompt for LiveBench Reasoning**

You are an expert in solving problems that require reasoning. You are given a QUESTION and a set of possible answers. Your job is to outline your step-by-step thought process for selecting the best answer.
QUESTION:
{question}
{answer_choices}
The output should be in JSON format: {{"reason": "<your reasoning>", "answer_id": "<an integer between 1 and {num_answer_choices}>"}}

---

For the LiveCodeBench TestOutputPred benchmark, we use the following multi-choice QA task prompt:

---

**Multi-choice QA Task Prompt for LiveCodeBench TestOutputPred**

You are an expert in solving problems that require reasoning. You are given a Problem and a set of possible answers. Your job is to outline your step-by-step thought process for selecting the best answer.
{problem}
{answer_choices}
The output should be in JSON format: {{"reason": "<your reasoning>", "answer_id": "<an integer between 1 and {num_answer_choices}>"}}

---

## C  Controlled Generation

For the NATURAL PLAN tasks, we use the controlled generation to output the solution in a structured format to improve the accuracy with Langfun. We use the following prompt to make the LLM output the final answer using the specified schema after chain-of-thought.

---

**Langfun Chain-of-Thought Answer Trigger**

Please think step by step to solve the task and then output a final solution using the specified schema.

---

We show the solution schema (Python class) definition for different datasets below.

```python
class Step(pg.Object):
  """One solution step."""

  city_name: Annotated[Optional[str], "The city name."]
  arrival_day: Annotated[Optional[int], "The day you arrive in the city."]
  departure_day: Annotated[
    Optional[int], "The day you depart from the city."
  ]
  duration: Annotated[
    Optional[int], "The number of days spent in the city."
  ]

class Solution(pg.Object):
  """The solution."""

  step_1: Step | None
  ...
  step_k: Step | None
```

Listing 1: Trip Planning solution class

```python

class Step(pg.Object):
  """One solution step."""

  location: Annotated[Optional[str], "The meeting location."]
  travel_time: Annotated[Optional[int], "The travel time in minutes."]
  arrival_time: Annotated[Optional[str], "The arrival time."]
  person: Annotated[Optional[str], "The person to meet at the location."]
  meeting_duration: Annotated[
    Optional[int], "The meeting duration in minutes."
  ]
  meeting_start_time: Annotated[Optional[str], "The meeting start time."]
  meeting_end_time: Annotated[Optional[str], "The meeting end time."]

class Solution(pg.Object):
  """The solution."""
```

```
18
19   step_1: Step | None
20   ...
21   step_k: Step | None
```

Listing 2: Meeting Planning solution class

# D Additional Results

## D.1 Cost Estimation using Number of API Calls

In this section, we show results when using the average number of API calls for measuring the computational cost. Figure 6 shows the scaling law curves where the x-axis is the average number of API calls and y-axis is the accuracy. The findings are the same as those where we use average number of output tokens to measure the cost.

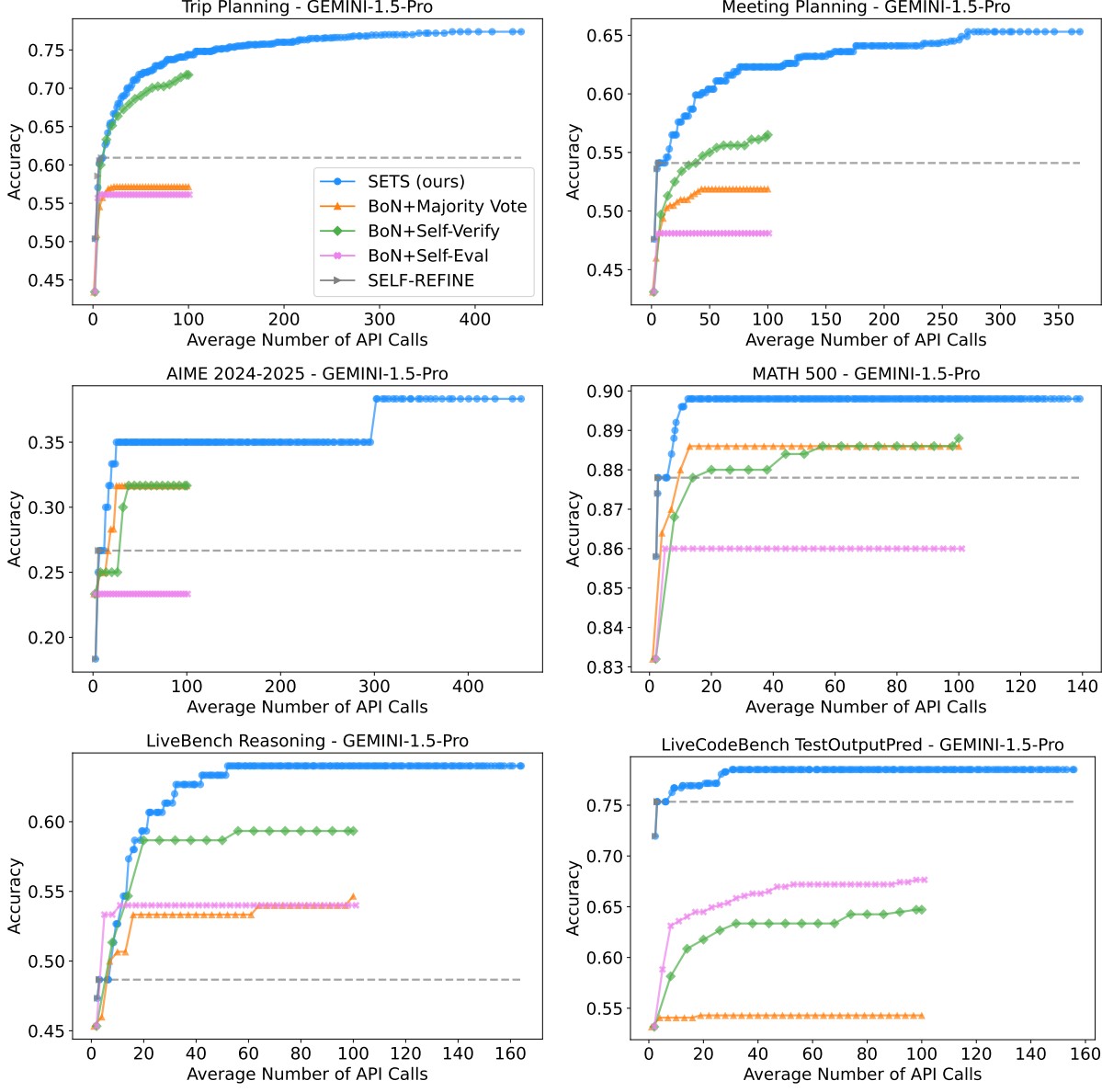

Figure 6: Scaling law curves where the x-axis is the average number of API calls and y-axis is the accuracy. Each point $(x, y)$ in the curve corresponds to a hyperparameter setting $\theta \in \Theta$. $y$ is the optimal performance at the cost budget $x = H(\theta)$ (refer to Section 4.1 for the details).

## D.2 Impact of Different LLMs

We apply SETS with Claude-3.5-Sonnet, GEMINI-2.5-Flash-Lite-Thinking, GEMINI-2.5-Flash, Qwen3-235B-A22B, and Qwen2.5-1.5B-Instruct. The results for these models are shown in Figure 7, Figure 8, Figure 9, Figure 10, and Figure 11, respectively. Slow inference speeds for Qwen3-235B-A22B and Qwen2.5-1.5B-Instruct restricted our experiments to Trip Planning (a 200-example subset), MATH 500, LiveBench Reasoning, and LiveCodeBench TestOutputPred. The findings are the same as those in Section 4.3: SETS outperforms the baselines on most of the cases with a few exceptions.

With the Qwen3-235B-A22B model, SETS consistently and significantly outperforms all baselines across four diverse benchmarks (planning, reasoning, math, and coding). This demonstrates that SETS remains highly effective even for open-weights models, improving performance despite less accurate initial outputs. These findings confirm that SETS is a robust technique for enhancing both state-of-the-art proprietary models and accessible open-source alternatives.

Conversely, results for the smaller Qwen2.5-1.5B-Instruct are more nuanced. SETS offers clear benefits on tasks where the model can generate and critique plausible solutions (e.g., LiveBench Reasoning and LiveCodeBench TestOutputPred). However, on MATH 500, it underperforms BoN+Majority Vote. On the complex Trip Planning benchmark, all methods failed entirely, as the base model lacked the fundamental capacity to generate valid solutions.

Finally, we observed that the GEMINI-2.5-Flash and GEMINI-2.5-Flash-Lite-Thinking models might fail to follow the specified instructions for Trip Planning and meeting planning, leading to incorrectly formatted responses. This formatting issue prevents the successful parsing of answers, resulting in a "None" value. For methods that use majority voting (SETS, BoN+Majority Vote, and BoN+Self-Verify), these "None" answers are excluded from the vote. Our results suggest that when the underlying language model has poor instruction-following abilities on a task, the proposed SETS method may not significantly outperform the baselines (e.g., using GEMINI-2.5-Flash-Lite-Thinking on Trip Planning).

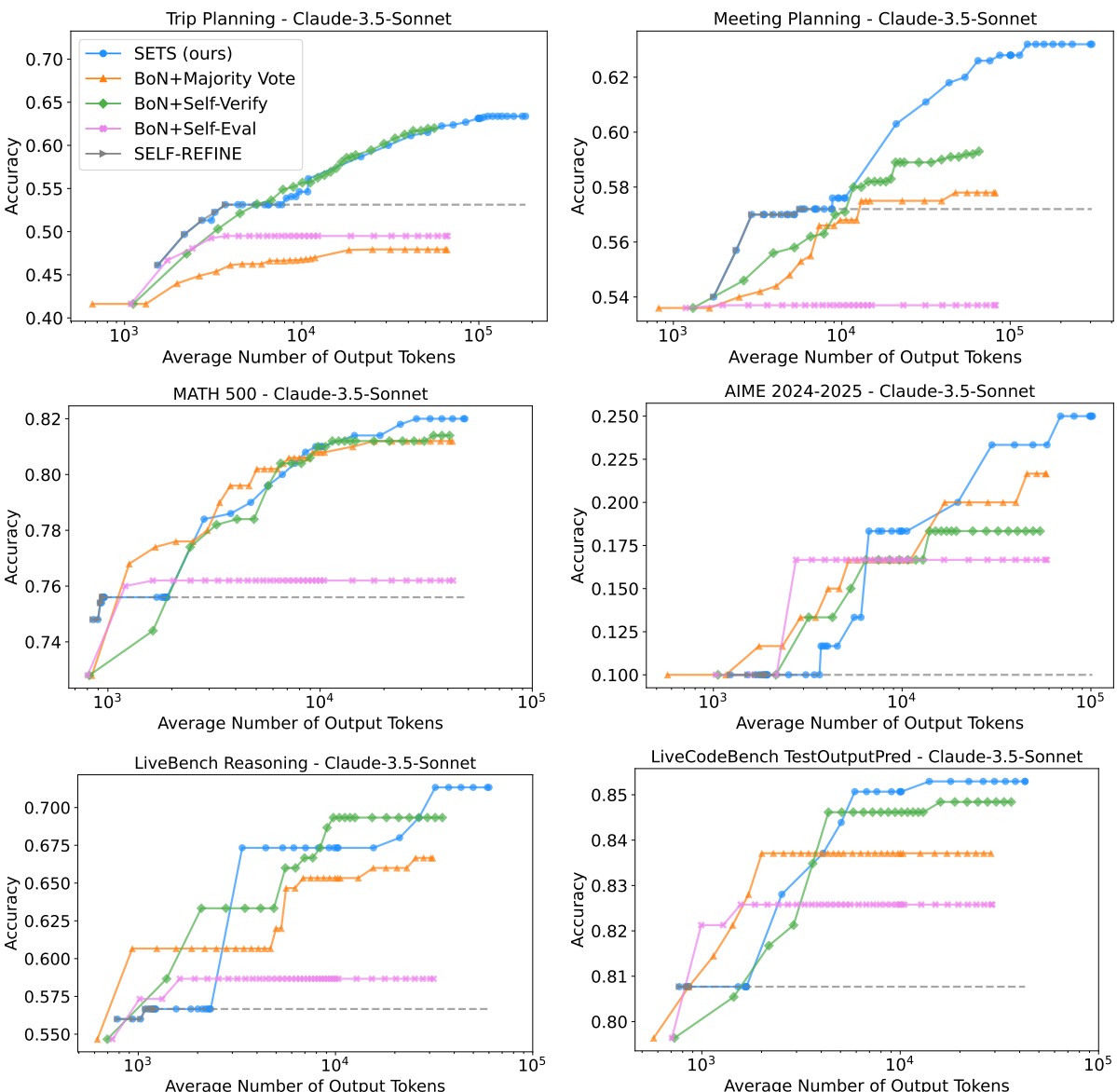

Figure 7: Scaling law curves for Claude-3.5-Sonnet.

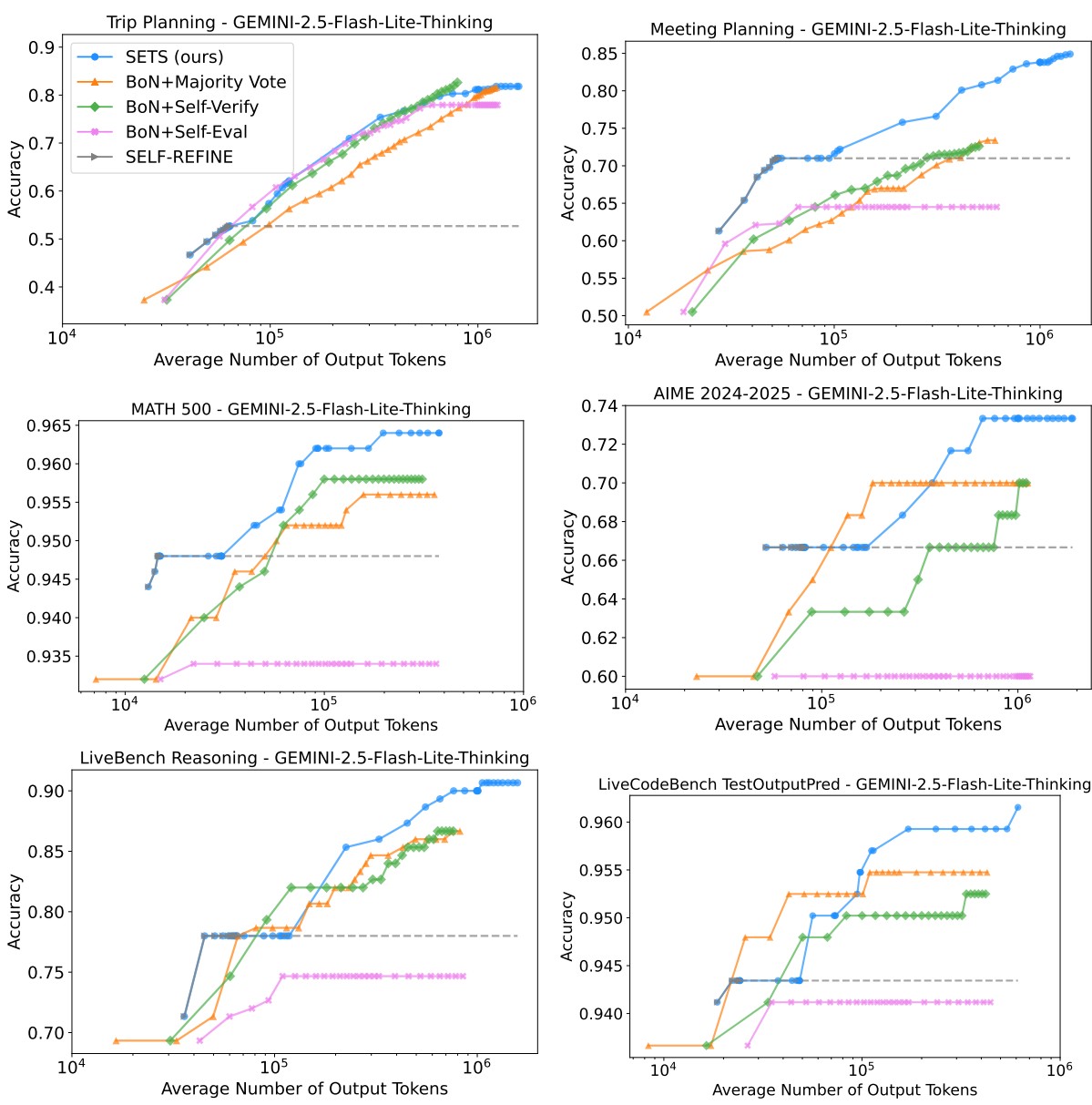

Figure 8: Scaling law curves for Gemini-2.5-Flash-Lite-Thinking.

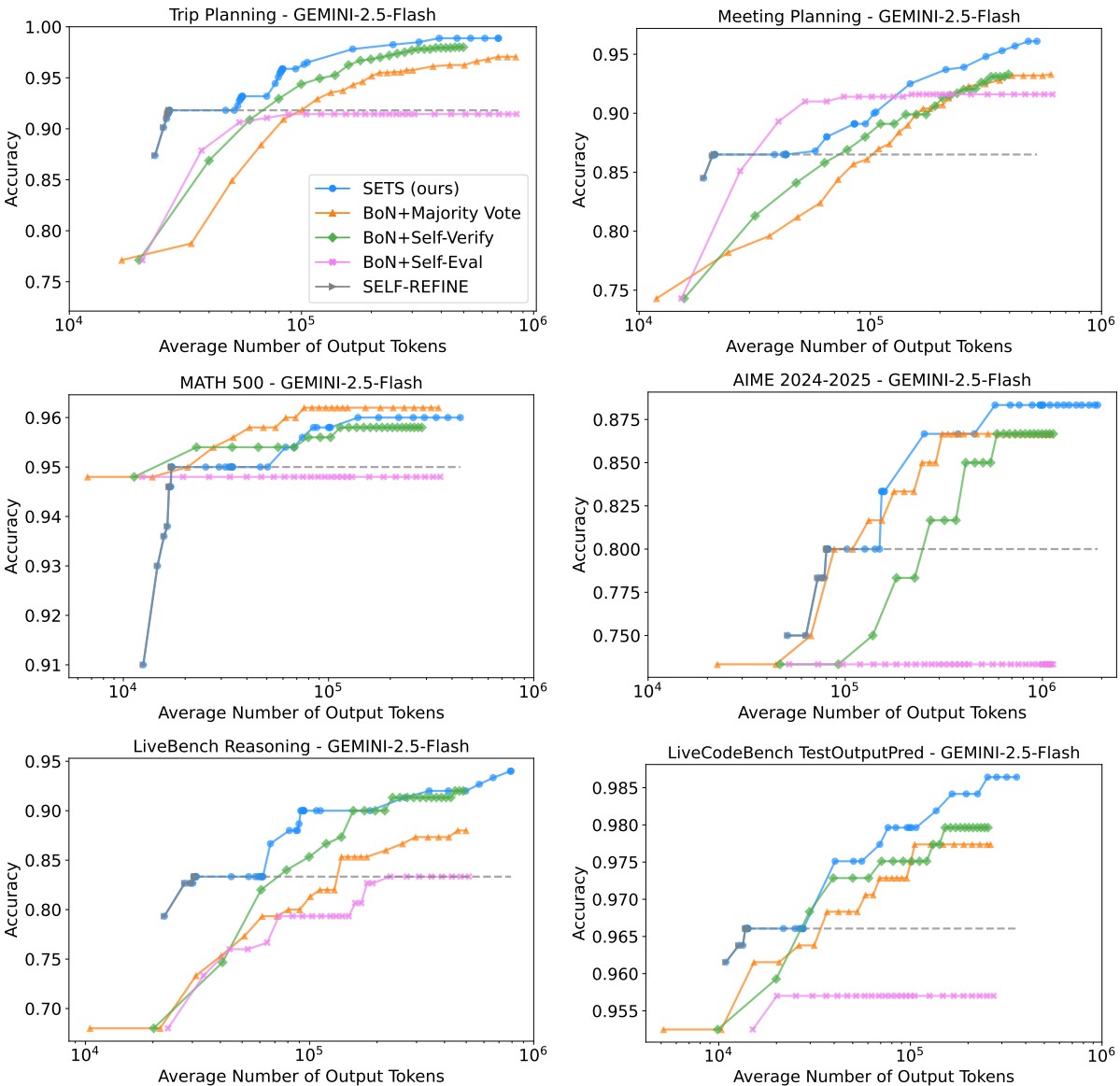

Figure 9: Scaling law curves for Gemini-2.5-Flash.

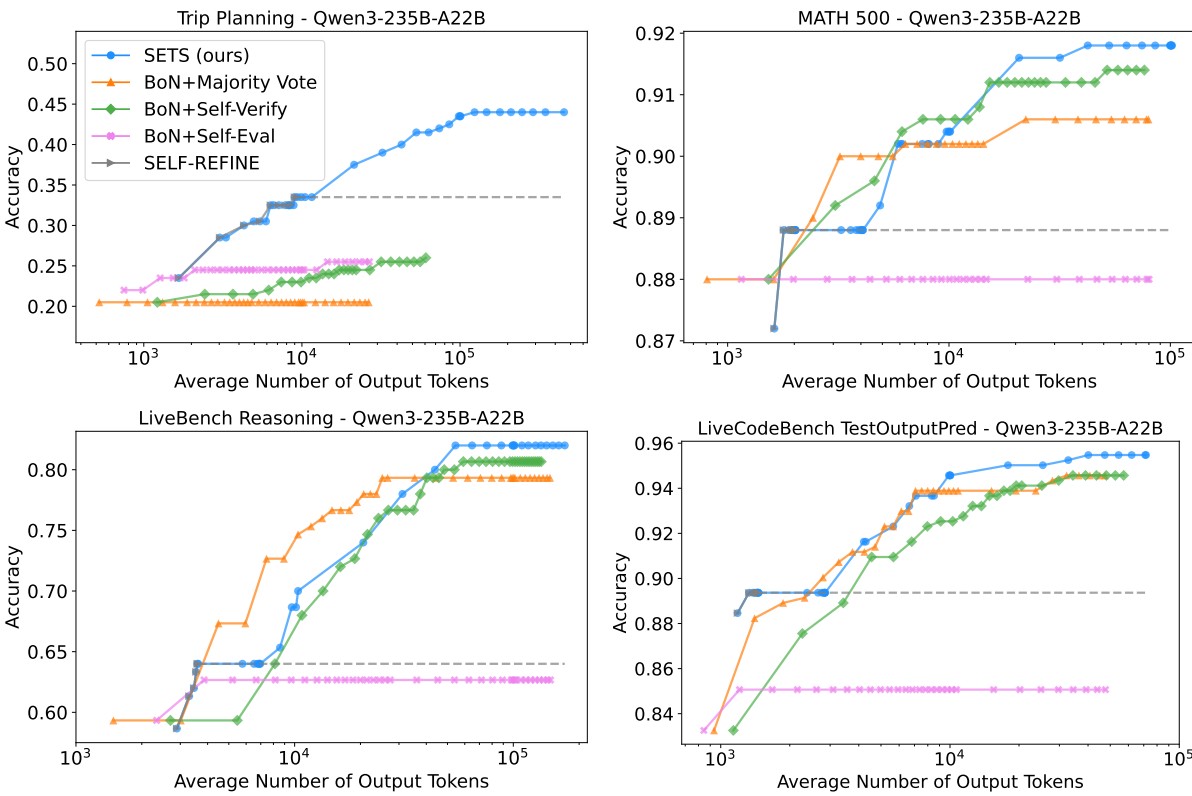

Figure 10: Scaling law curves for Qwen3-235B-A22B.

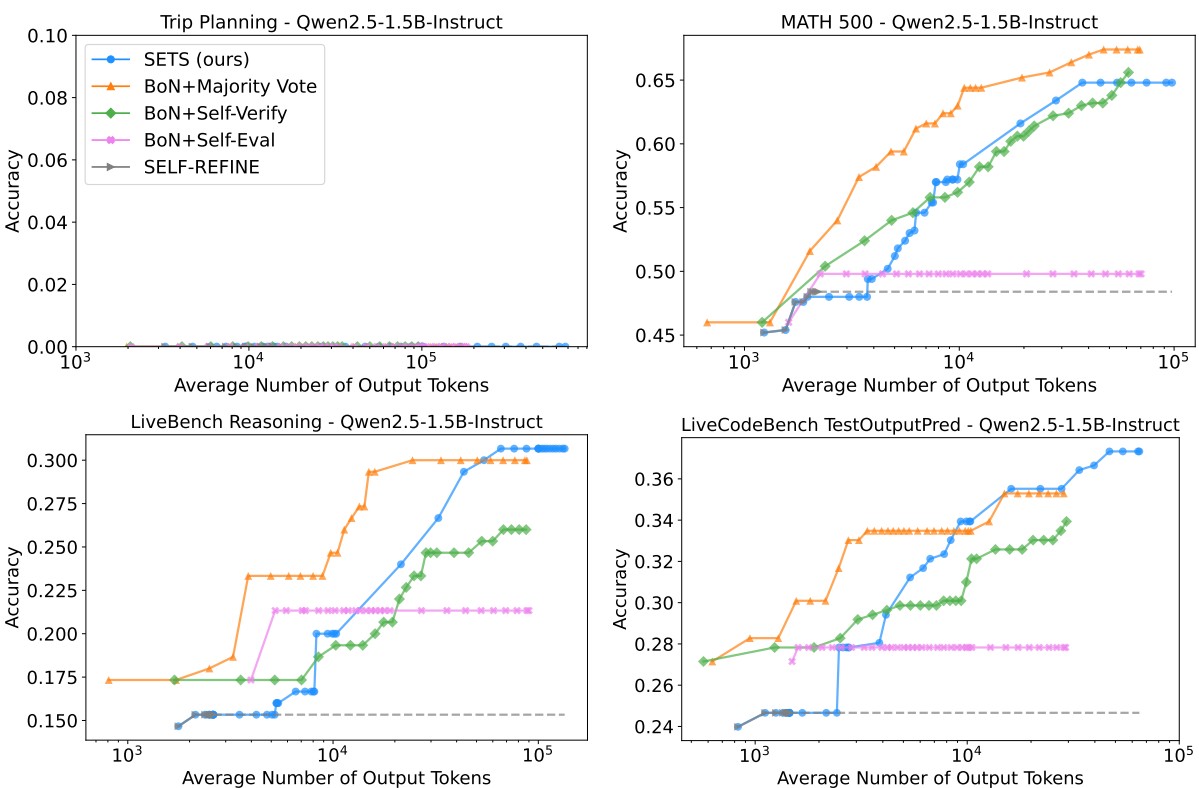

Figure 11: Scaling law curves for Qwen2.5-1.5B-Instruct.

### D.3 Evaluating Self-Verification Performance

We study whether more self-verification samples will improve the self-verification performance. We ask the LLM to self-verify its own proposed solution (sampled with temperature= 0) multiple times and define the verification score as the fraction of times that the solution is verified as correct. We then use the AUROC metric to measure the correlation between the verification score and the correctness of the proposed solution, which can reflect the self-verification performance. The results in Figure 12 show that increasing the number of self-verification samples lead to better self-verification performance, but the performance typically saturates quickly. These results justify the design of the proposed method SETS: adding the dimension of the number of samples $m$ allows the LLM to self-verify the same solution multiple times, which can improve the self-verification performance.

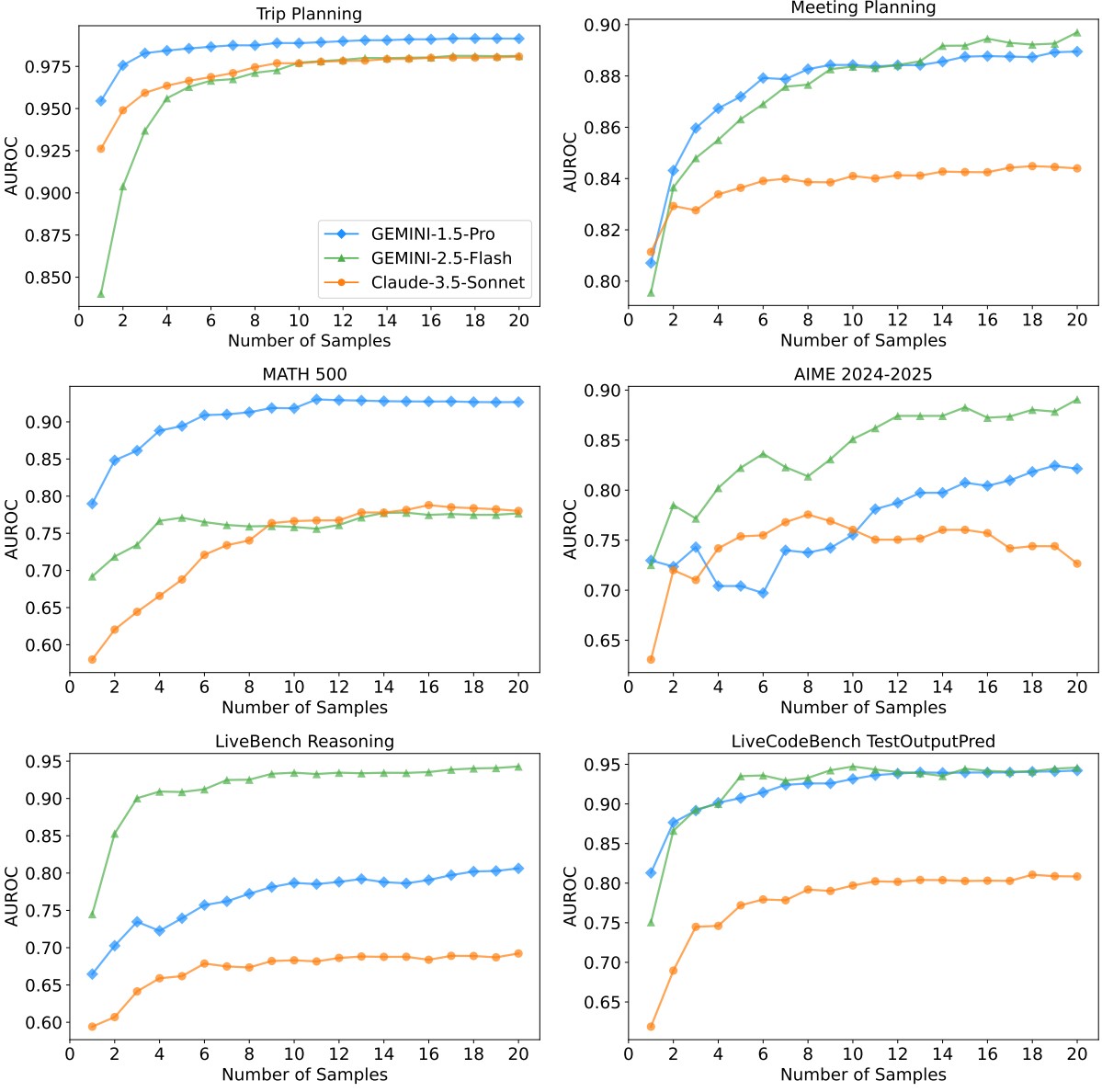

Figure 12: Evaluate the self-verification performance of different models as we increase the number of self-verification samples.

### D.4 The Effect of Self-Correction Rounds

In Figure 13, we show the results for studying the effect of self-correction rounds on MATH 500, AIME 2024-2025, LiveBench Reasoning and LiveCodeBench TestOutputPred datasets. The findings are the same as those in Section 4.3.

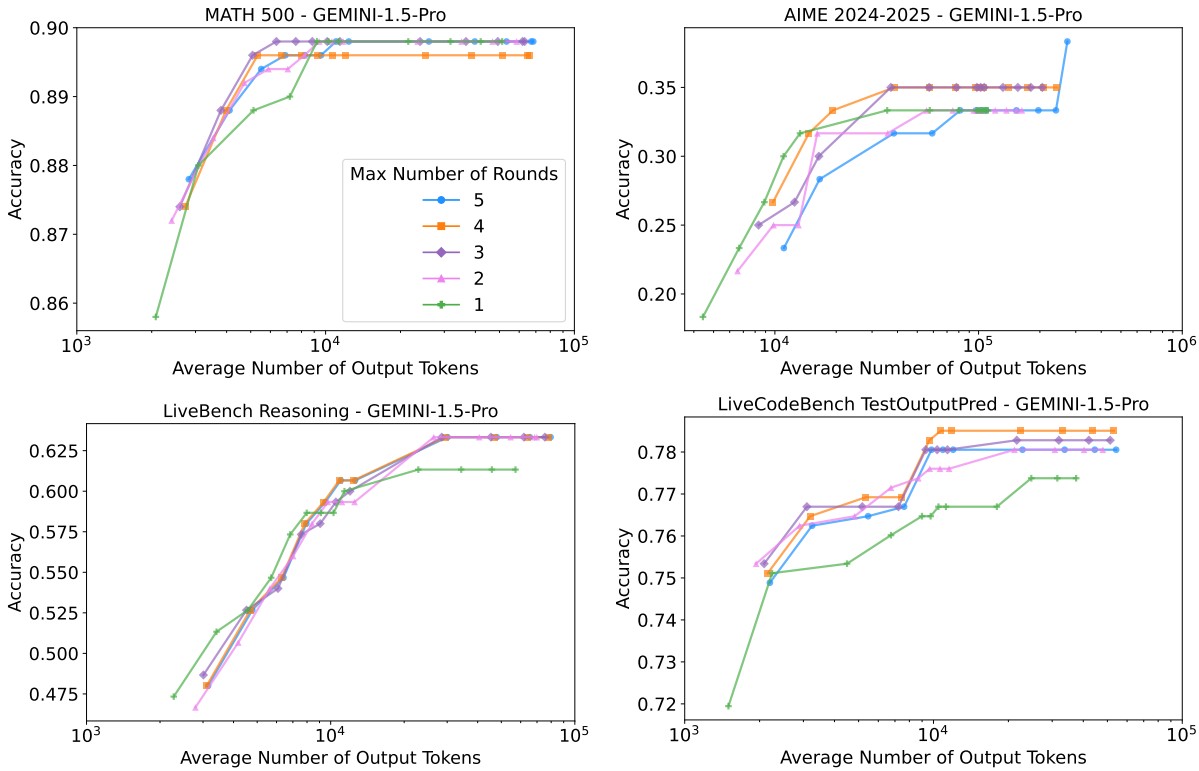

Figure 13: The effect of allocating more compute to self-verification and self-correction for SETS (controlled by Max Number of Rounds) given a fixed computational budget (measured by Average Number of Output Tokens).

### D.5    The Effect of Temperature for SETS

We study how the temperature used for the three core operations (Sampling, Self-Verify, and Self-Correct) affects the performance of SETS. We consider two configurations: (1) using a temperature of 0.7 for all three operations (our default setting), and (2) using a temperature of 0.7 for Sampling, but a temperature of 0.0 (greedy decoding) for Self-Verify and Self-Correct. The results on the Meeting Planning and LiveBench Reasoning benchmarks are shown in Figure 14. The findings are the same as those in Section 4.3: our default setting generally achieves better performance across different benchmarks.

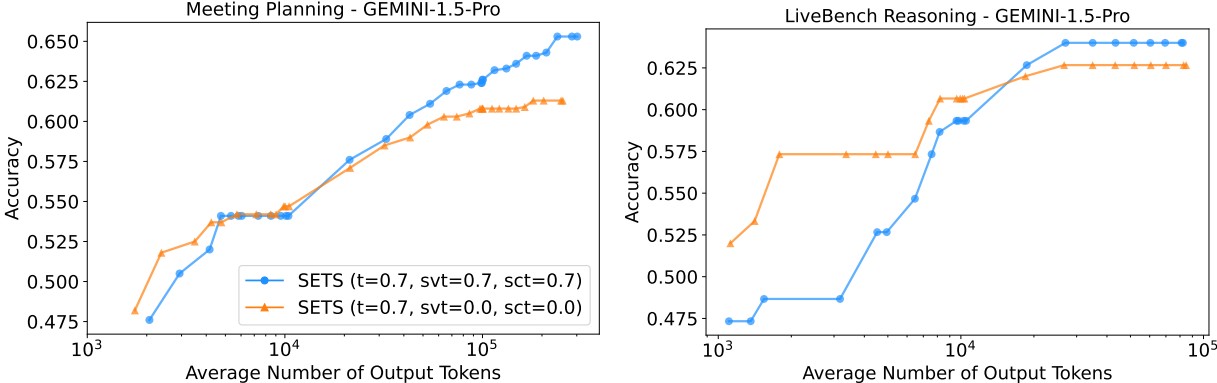

Figure 14: The effect of different temperature settings for SETS. t, svt and sct are temperature parameters for the Sampling, Self-Verify and Self-Correct operations respectively.

## D.6 Performance under the Oracle Setting

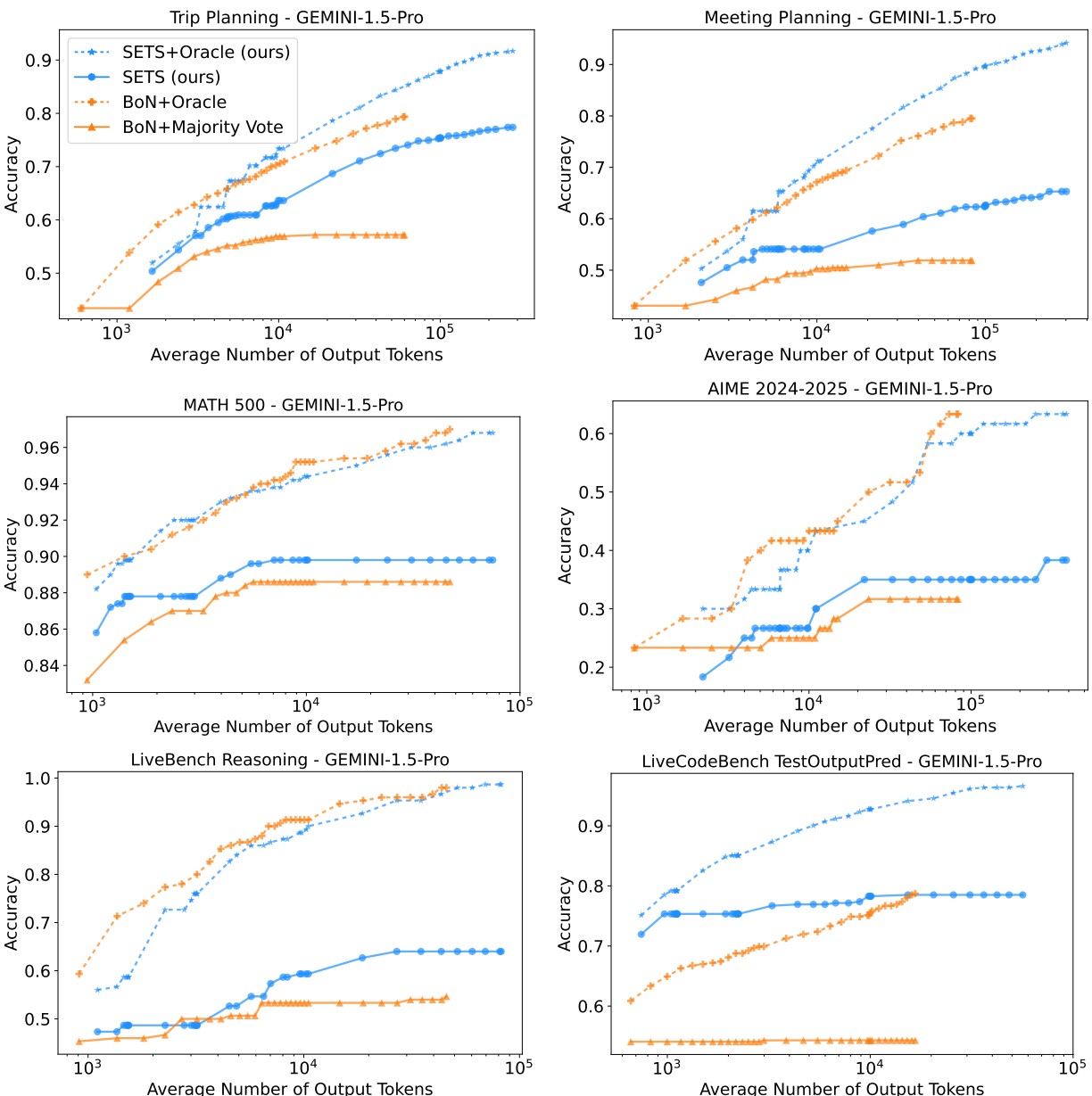

Figure 15: Scaling law curves under the oracle setting where the x-axis is the average number of output tokens and y-axis is the accuracy. Each point $(x, y)$ in the curve corresponds to a hyperparameter setting $\theta \in \Theta$. $y$ is the optimal performance at the cost budget $x = H(\theta)$ (see Section 4.1 for details). We subsample the points (up to 8 within every x-tick interval) to make the markers less crowded.

We compare the proposed method SETS with the Best-of-N method under the oracle setting where the final solution is selected using ground-truth reference. Note that this oracle setting is not feasible in practice as it depends on ground-truth labels. We consider the following two oracle methods:

- **BoN+Oracle**: We first sample $m$ solutions and then select the final solution using ground-truth reference. If all sampled solutions are incorrect, we select the first sampled solution.

- **SETS+Oracle**: We select the final solution among the solutions generated by SETS (up to $m \cdot (n+1)$ solutions). If all solutions generated by SETS are incorrect, we select the first sampled solution.

We perform experiments for GEMINI-1.5-Pro and the results are shown in Figure 15. We can see that SETS with oracle selection has a marked advantage over BoN with oracle selection on Trip Planning, Meeting Planning and LiveCodeBench TestOutputPred while the advantage is less pronounced on the other tasks. This may suggest that SETS is more effective on tasks with larger and more complex solution space. Notably, on LiveCodeBench TestOutputPred, SETS outperforms the performance of BoN+Oracle that uses ground-truth labels for solution selection. This indicates that when the LLM possesses strong self-verification and self-correction capabilities, SETS provides an efficient way to scale test-time compute and thus enhance overall accuracy.

### D.7 The Impact of Task Difficulty

The NATURAL PLAN datasets provide a controlled variable (e.g., the number of people or the number of cities) that indicates the difficulty level of each task. We utilize this controlled variable to study the performance of SETS on the easy and hard tasks. For Trip Planning, we treat a task with no more than 6 cities as an easy task, and otherwise a hard task. For Meeting Planning, we treat a task with no greater than 5 people as an easy task, and otherwise a hard task. Figure 16 shows that SETS significantly outperforms the baselines on both easy and hard tasks. On hard tasks, SETS also brings significant accuracy gains and can achieve higher accuracy if more test-time compute is used.

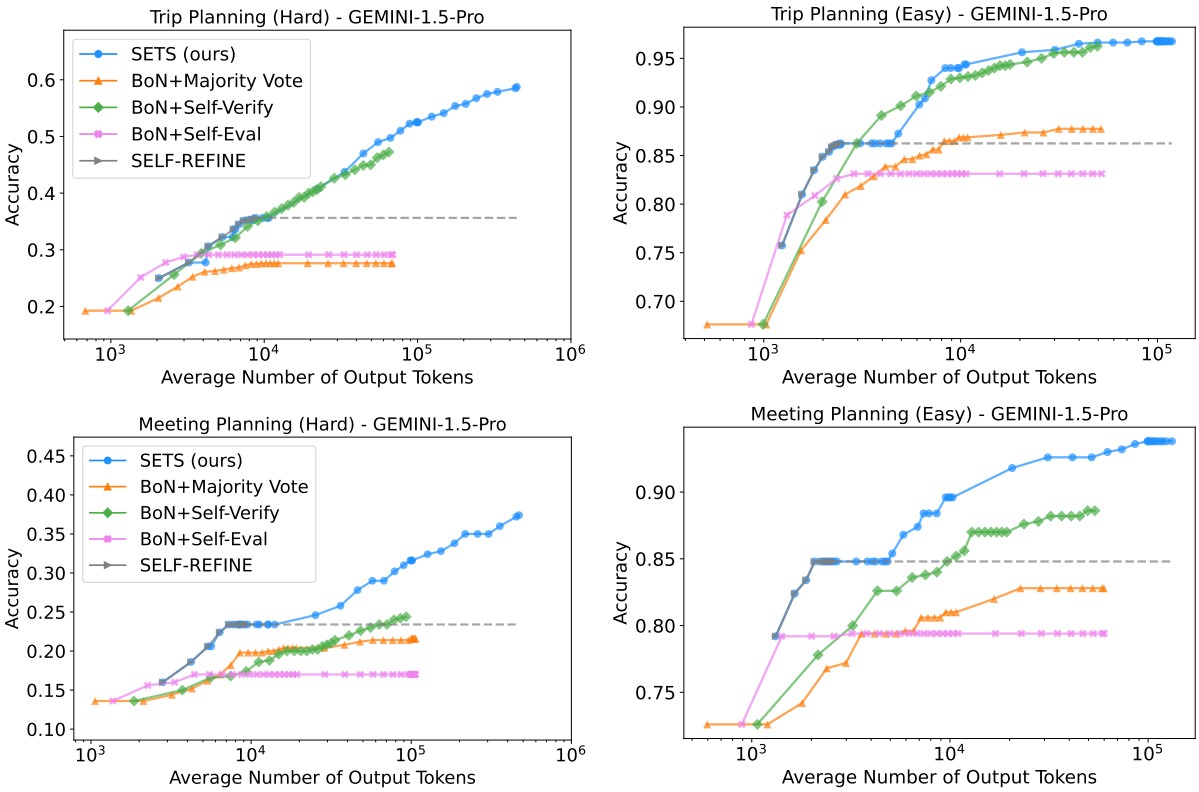

Figure 16: Estimated scaling law curves for Hard and Easy tasks obtained with SETS vs. baselines.

### D.8 Performance under fixed hyperparameters

We evaluate SETS and baseline methods using fixed hyperparameters. For BoN+Majority Vote and BoN+Self-Eval, we set $m = 100$. For BoN+Self-Verify, we set $m = 50$. For SELF-REFINE, we set $n = 5$. For SETS, we set $m = 20$ and $n = 3$. We repeat each experiment three times and report the mean and standard deviation for all metrics. As shown in Table 4, SETS generally demonstrates significantly superior performance over the Best-of-N (BoN) baselines when operating under comparable computational budgets, measured by the average number of output tokens. While the SELF-REFINE method consumes considerably less computational resources than SETS, its accuracy is substantially lower.

| Dataset | Method | Accuracy (%) ↑ | Avg. # Output Tokens | Avg. # API Calls |
|---|---|---|---|---|
| Trip Planning | BoN+Majority Vote | 49.9±6.2 | 35218±21558 | 100±0 |
| | BoN+Self-Verify | 60.6±9.7 | 44844±10756 | 100±0 |
| | BoN+Self-Eval | 40.8±1.2 | 60469±10 | 101±0 |
| | SELF-REFINE | 59.5±0.0 | 4233±0 | 6±0 |
| | SETS (ours) | **68.5**±3.0 | 48364±10857 | 87±0 |
| Meeting Planning | BoN+Majority Vote | 48.8±2.3 | 56422±23057 | 100±0 |
| | BoN+Self-Verify | 55.7±0.9 | 73323±39 | 100±0 |
| | BoN+Self-Eval | 40.7±0.5 | 83273±2 | 101±0 |
| | SELF-REFINE | 54.1±0.0 | 4746±0 | 5±0 |
| | SETS (ours) | **59.6**±1.4 | 62899±9196 | 86±0 |
| MATH 500 | BoN+Majority Vote | 87.8±0.2 | 47026±21 | 100±0 |
| | BoN+Self-Verify | 88.7±0.1 | 46997±21 | 100±0 |
| | BoN+Self-Eval | 75.9±0.8 | 47407±1 | 101±0 |
| | SELF-REFINE | 87.8±0.0 | 1412±0 | 3±0 |
| | SETS (ours) | **89.3**±0.1 | 25392±147 | 50±0 |
| AIME 2024-2025 | BoN+Majority Vote | 32.2±1.0 | 82942±221 | 100±0 |
| | BoN+Self-Verify | 28.3±1.7 | 84469±130 | 100±0 |
| | BoN+Self-Eval | 8.9±3.5 | 83759±17 | 101±0 |
| | SELF-REFINE | 23.3±0.0 | 5232±0 | 6±0 |
| | SETS (ours) | **33.3**±0.0 | 81446±826 | 99±1 |
| LiveBench Reasoning | BoN+Majority Vote | 52.2±1.4 | 45711±172 | 100±0 |
| | BoN+Self-Verify | 58.7±1.2 | 50230±120 | 100±0 |
| | BoN+Self-Eval | 47.3±4.4 | 46342±2 | 101±0 |
| | SELF-REFINE | 48.0±0.0 | 1526±0 | 3±0 |
| | SETS (ours) | **60.7**±1.3 | 29954±119 | 60±0 |
| LiveCodeBench TestOutputPred | BoN+Majority Vote | 54.3±0.2 | 16546±129 | 100±0 |
| | BoN+Self-Verify | 64.9±0.5 | 29789±2 | 100±0 |
| | BoN+Self-Eval | 68.8±1.1 | 17126±23 | 101±0 |
| | SELF-REFINE | 74.9±0.0 | 1095±0 | 3±0 |
| | SETS (ours) | **77.8**±0.2 | 20616±71 | 59±0 |

Table 4: Performance under fixed hyper-parameters with GEMINI-1.5-Pro. We show the mean and standard deviation of the metrics (mean±std). **Bold** numbers are superior results.

### D.9 Financial Cost Estimation

In this section, we show results when using the average price for measuring the cost. Figure 17 shows the scaling law curves where the x-axis is the average price and y-axis is the accuracy. The findings are the same as those where we use average number of output tokens to measure the cost.

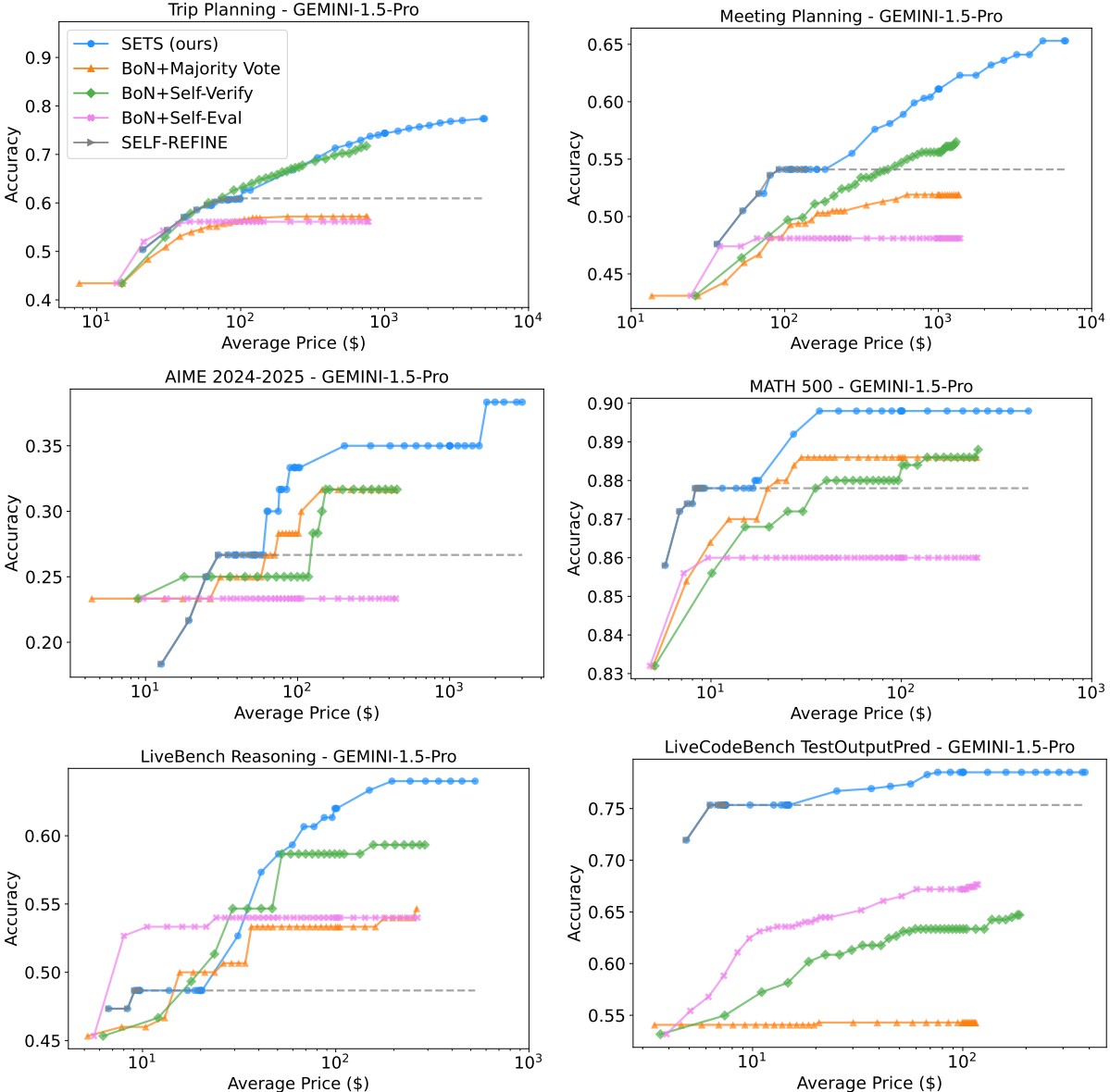

Figure 17: Scaling law curves where the x-axis is the average price and y-axis is the accuracy. Each point $(x, y)$ in the curve corresponds to a hyperparameter setting $\theta \in \Theta$. $y$ is the optimal performance at the cost budget $x = H(\theta)$ (refer to Section 4.1 for the details).

### D.10 Non-thinking Mode with SETS vs. Thinking Mode

To clarify the role of test-time scaling relative to a model's intrinsic power, we compared the non-thinking mode using SETS against the more powerful thinking mode with BoN+Majority Vote, under a fixed token budget. The results in Table 5 show that SETS does not bridge the fundamental capability gap between the two modes. This finding highlights that SETS functions as capability amplifiers, not creators; they enhance a model's existing reasoning rather than acting as a substitute for it.

The true utility of SETS is therefore realized when maximizing a given model's potential. Indeed, as demonstrated in Table 5, applying SETS to the thinking mode itself yields substantial performance gains over the thinking mode with BoN+Majority Vote. This confirms that the practical value of SETS lies in pushing the performance ceiling of a chosen model – including state-of-the-art ones – to its absolute limit.

| Dataset | Model | Method | Accuracy (%) ↑ | |
| --- | --- | --- | --- | --- |
| | | | Budget=$10^5$ | Budget=$10^6$ |
| Trip Planning | GEMINI-2.5-Flash-Lite (non-thinking) | SETS | 54.31 | 64.56 |
| | GEMINI-2.5-Flash-Lite-Thinking | BoN+Majority Vote | 53.06 | 80.00 |
| | GEMINI-2.5-Flash-Lite-Thinking | SETS | **57.38** | **81.12** |
| Meeting Planning | GEMINI-2.5-Flash-Lite (non-thinking) | SETS | 64.20 | 70.50 |
| | GEMINI-2.5-Flash-Lite-Thinking | BoN+Majority Vote | 62.70 | 73.40 |
| | GEMINI-2.5-Flash-Lite-Thinking | SETS | **71.60** | **83.80** |
| MATH 500 | GEMINI-2.5-Flash-Lite (non-thinking) | SETS | 94.20 | 94.20 |
| | GEMINI-2.5-Flash-Lite-Thinking | BoN+Majority Vote | 95.20 | 95.60 |
| | GEMINI-2.5-Flash-Lite-Thinking | SETS | **96.20** | **96.40** |
| AIME 2024-2025 | GEMINI-2.5-Flash-Lite (non-thinking) | SETS | 60.00 | 66.67 |
| | GEMINI-2.5-Flash-Lite-Thinking | BoN+Majority Vote | 65.00 | 70.00 |
| | GEMINI-2.5-Flash-Lite-Thinking | SETS | **66.67** | **73.33** |
| LiveBench Reasoning | GEMINI-2.5-Flash-Lite (non-thinking) | SETS | 78.00 | 82.67 |
| | GEMINI-2.5-Flash-Lite-Thinking | BoN+Majority Vote | **78.67** | 86.67 |
| | GEMINI-2.5-Flash-Lite-Thinking | SETS | 78.00 | **90.00** |
| LiveCodeBench TestOutputPred | GEMINI-2.5-Flash-Lite (non-thinking) | SETS | 94.12 | 94.12 |
| | GEMINI-2.5-Flash-Lite-Thinking | BoN+Majority Vote | 95.25 | 95.48 |
| | GEMINI-2.5-Flash-Lite-Thinking | SETS | **95.48** | **96.15** |

Table 5: Comparison of Thinking Mode and Non-Thinking Mode with SETS under the same output token budget. **Bold** numbers are superior results.

### D.11 SETS with Confidence-weighted Voting

To explore a more sophisticated aggregation strategy, we evaluated SETS with confidence-weighted voting. In this approach, each candidate solution is weighted by a confidence score, defined as the proportion of times it is verified as correct during the scaling process.

As shown in Figure 18, this method generally enhances performance over standard majority voting, achieving superior results on most benchmarks (Trip Planning, MATH 500, LiveBench Reasoning, and LiveCodeBench TestOutputPred). However, this improvement is not universal. On certain tasks, such as Meeting Planning and AIME 2024-2025, the simpler majority vote remains more effective, suggesting that the reliability of self-verification scores as a confidence heuristic can be task-dependent.

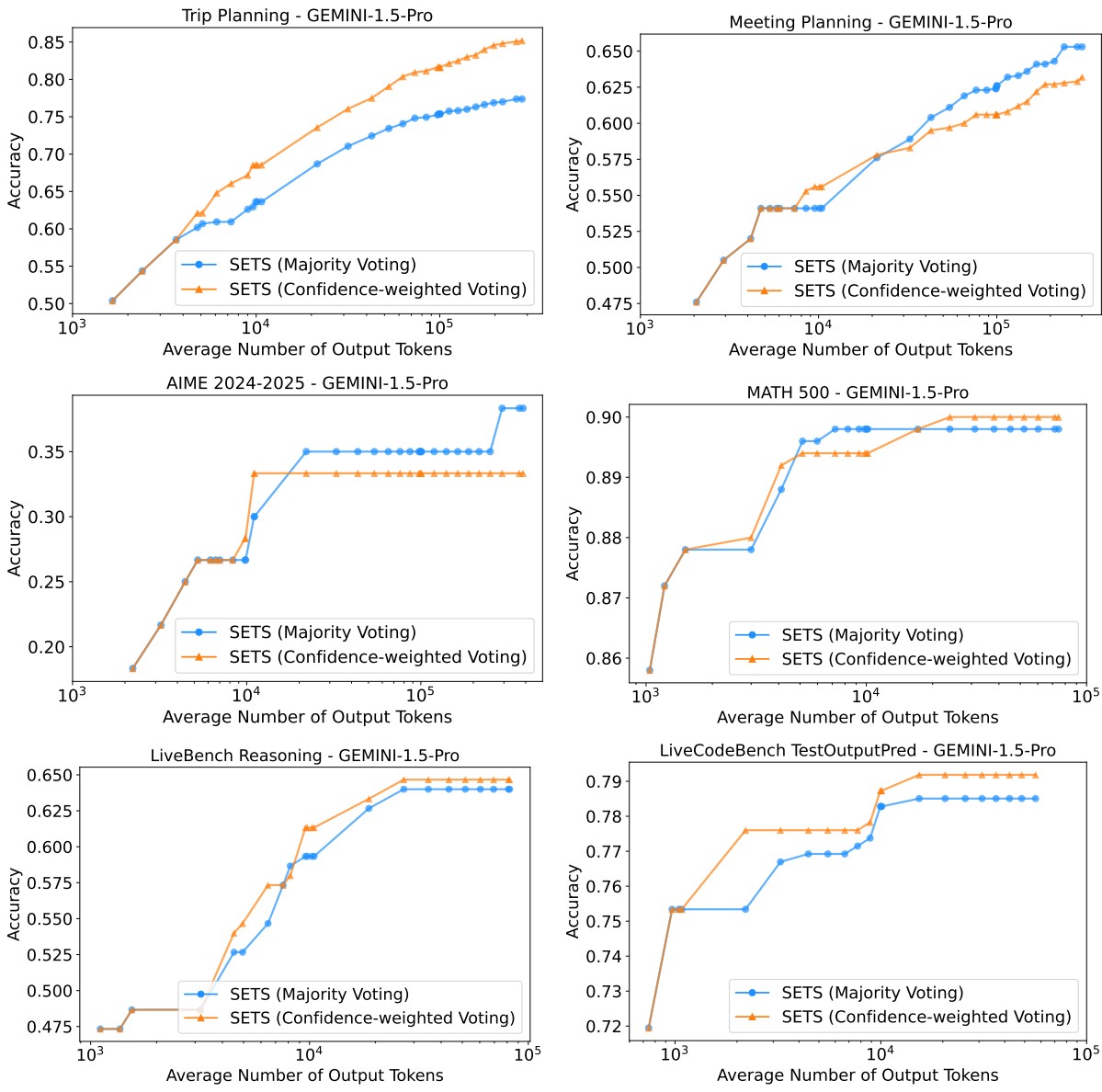

Figure 18: The effect of confidence-weighted voting on SETS performance.

### D.12 Failure Modes of Self-Verification

This section provides a qualitative analysis of self-verification's failure modes. Our review indicates that these failures, with incorrect verification steps highlighted in red, primarily stem from model hallucinations, where the model generates factually incorrect or nonsensical reasoning.

---

**Trip Planning (False Negative)**

PROBLEM:
You plan to visit 3 European cities for 10 days in total. You only take direct flights to commute between cities. On the last day of your visit to each city, you can take a direct flight to the next city and arrive on the same day. Both the day you arrive and the day you depart count toward the total number of days spent in each city. You want to spend 7 days in Santorini. You want to spend 3 days in Amsterdam. You would like to visit Lyon for 2 days. From day 1 to day 2, there is a annual show you want to attend in Lyon.
Here are the cities that have direct flights: Lyon and Amsterdam, Amsterdam and Santorini.
Find a trip plan of visiting the cities for 10 days by taking direct flights to commute between them.
PROPOSED SOLUTION:
Solution( step_1 = Step( city_name = 'Lyon', arrival_day = 1, departure_day = 2, duration = 2 ), step_2 = Step( city_name = 'Amsterdam', arrival_day = 2, departure_day = 4, duration = 3 ), step_3 = Step( city_name = 'Santorini', arrival_day = 4, departure_day = 10, duration = 7 ) )
SELF-VERIFICATION:
1. **Constraints:** * **Total trip duration:** 10 days. * **Number of cities:** 3 * **Direct flights only:** Travel between cities must be via direct flights. * **Same day arrival/departure:** Flights can be taken on the last day of the stay in a city, arriving at the next city on the same day. * **Inclusive days:** Arrival and departure days count towards the total days spent in a city. * **Santorini duration:** 7 days. * **Amsterdam duration:** 3 days. * **Lyon duration:** 2 days. * **Lyon event:** Visit Lyon between day 1 and day 2. * **Direct flight availability:** Direct flights exist between Lyon and Amsterdam, and Amsterdam and Santorini.
2. **Verification:**
* **Total trip duration:** The trip starts on day 1 in Lyon and ends on day 10 in Santorini, totaling 10 days. *Correct* * **Number of cities:** The itinerary includes Lyon, Amsterdam, and Santorini. *Correct* * **Direct flights only:** The proposed flights (Lyon to Amsterdam and Amsterdam to Santorini) are listed as available direct flights. *Correct* * **Same day arrival/departure:** The departure day from Lyon (day 2) is the same as the arrival day in Amsterdam. The departure day from Amsterdam (day 4) is the same as the arrival day in Santorini. *Correct* * **Inclusive days:** * Lyon: Arrival day 1 and departure day 2, making 2 days. *Correct* * Amsterdam: Arrival day 2 and departure day 4, making 3 days. *Correct* * Santorini: Arrival day 4 and departure day 10, making 7 days. *Correct* * **Santorini duration:** The itinerary allocates 7 days to Santorini (day 4 to day 10). *Correct* * **Amsterdam duration:** The itinerary allocates 3 days to Amsterdam (day 2 to day 4). *Correct* * **Lyon duration:** The itinerary allocates 2 days to Lyon (day 1 to day 2). *Correct* * **Lyon event:** The itinerary includes Lyon on days 1 and 2, satisfying the requirement to be in Lyon for the event. *Correct* * **Direct flight availability:** The proposed route uses only available direct flights. *Correct*
3. The proposed solution is incorrect. The proposed solution allocates 3 days to Amsterdam (arrival on day 2, departure on day 4). However, it should be 2 days and 4, totaling 3 days. Thus, the proposed solution is incorrect.

---

## Trip Planning (False Positive)

PROBLEM:
You plan to visit 4 European cities for 16 days in total. You only take direct flights to commute between cities. On the last day of your visit to each city, you can take a direct flight to the next city and arrive on the same day. Both the day you arrive and the day you depart count toward the total number of days spent in each city. You want to spend 5 days in Paris. You want to meet a friend in Paris between day 1 and day 5. You plan to stay in Santorini for 6 days. You want to spend 2 days in Stockholm. You would like to visit Athens for 6 days. You are going to attend a wedding in Athens between day 5 and day 10. Here are the cities that have direct flights: Paris and Stockholm, Paris and Athens, Athens and Santorini, Athens and Stockholm, from Stockholm to Santorini. Find a trip plan of visiting the cities for 16 days by taking direct flights to commute between them.

PROPOSED SOLUTION:
Solution( step_1 = Step( city_name = 'Paris', arrival_day = 1, departure_day = 5, duration = 5 ), step_2 = Step( city_name = 'Athens', arrival_day = 5, departure_day = 10, duration = 6 ), step_3 = Step( city_name = 'Santorini', arrival_day = 10, departure_day = 15, duration = 6 ), step_4 = Step( city_name = 'Stockholm', arrival_day = 15, departure_day = 16, duration = 2 ) )

SELF-VERIFICATION:
1. **Constraints:** * **Total Duration:** 16 days. * **Cities:** Paris, Santorini, Stockholm, Athens. * **Direct Flights:** Only direct flights are allowed between cities. * **Paris:** 5 days, between day 1 and day 5. * **Santorini:** 6 days. * **Stockholm:** 2 days. * **Athens:** 6 days, between day 5 and day 10. * **Flights:** Paris and Stockholm, Paris and Athens, Athens and Santorini, Athens and Stockholm, Stockholm to Santorini. 2. **Verification:** * **Total Duration:** The trip spans from day 1 to day 16, which is 16 days. *Correct.* * **Cities:** The solution includes Paris, Santorini, Stockholm, and Athens. *Correct.* * **Direct Flights:** * Paris to Athens: *Correct.* * Athens to Santorini: *Correct.* * Santorini to Stockholm: *Correct.* * **Paris:** The solution has Paris from day 1 to day 5 (5 days), which is within the specified range. *Correct.* * **Santorini:** The solution has Santorini from day 10 to day 15 (6 days). *Correct.* * **Stockholm:** The solution has Stockholm from day 15 to day 16 (2 days). *Correct.* * **Athens:** The solution has Athens from day 5 to day 10 (6 days), which is within the specified range. *Correct.* The proposed solution is correct

# E Examples for Three Core Operations

In this section, we show the detailed responses for the three core operations (Sampling, Self-Verify, and Self-Correct) employed within SETS on a problem from MATH 500 with GEMINI-1.5-Pro.

---

**Problem and Ground Truth Answer**

PROBLEM: What is the least positive integer multiple of 30 that can be written with only the digits 0 and 2?
Ground Truth Answer: 2220

---

**Sampling Response**

Let $n$ be the least positive integer multiple of 30 that can be written with only the digits 0 and 2. Since $n$ is a multiple of 30, it must be a multiple of 10, meaning its last digit must be 0. Also, since $n$ is a multiple of 30, it must be a multiple of 3, meaning the sum of its digits must be a multiple of 3. Since $n$ can only contain the digits 0 and 2, the sum of the digits must be a multiple of both 2 and 3. The smallest possible number with only 0 and 2 as digits and ending in 0 is 20, which is not divisible by 3. The next smallest number is 200, which is not divisible by 3. The next smallest number is 2020, the sum of the digits is $2 + 0 + 2 + 0 = 4$, which is not divisible by 3. The next smallest number is 2000, the sum of the digits is $2 + 0 + 0 + 0 = 2$, which is not divisible by 3. The next smallest number is 20020, the sum of digits is $2 + 0 + 0 + 2 + 0 = 4$, not divisible by 3. The next smallest number is 20200, the sum of digits is $2 + 0 + 2 + 0 + 0 = 4$, not divisible by 3. The next smallest number is 202020, the sum of digits is $2 + 0 + 2 + 0 + 2 + 0 = 6$, which is divisible by 3. Since it ends in 0, it is divisible by 10. So it is divisible by 30. $202020 = 30 \cdot 6734$ $202020 \div 30 = 6734$ Therefore, the least positive integer multiple of 30 that can be written with only the digits 0 and 2 is 202020.
Final Answer: The final answer is $\boxed{202020}$

---

**Self-Verify Response**

1. **Conditions and Constraints:**
* **Multiple of 30:** The number must be divisible by both 3 and 10. * **Digits 0 and 2 only:** The number can only contain the digits 0 and 2. * **Least positive integer:** We are looking for the smallest positive integer satisfying the above conditions. * **Positive:** The number must be greater than 0. * **Integer:** The number must be a whole number.
2. **Verification:**
* **Digits 0 and 2 only:** The proposed answer 202020 consists only of the digits 0 and 2. This condition is satisfied. * **Positive:** 202020 is greater than 0. This condition is satisfied. * **Integer:** 202020 is a whole number. This condition is satisfied. * **Divisible by 10:** A number is divisible by 10 if its last digit is 0. The last digit of 202020 is 0. This condition is satisfied. * **Divisible by 3:** A number is divisible by 3 if the sum of its digits is divisible by 3. The sum of the digits of 202020 is $2 + 0 + 2 + 0 + 2 + 0 = 6$. Since 6 is divisible by 3, 202020 is divisible by 3. This condition is satisfied. * **Least positive integer:** Let's consider smaller numbers formed with 0s and 2s. We need a number divisible by 10, so it must end in 0.
* 20: Divisible by 10, but not by 3 (2+0=2). * 200: Divisible by 10, but not by 3 (2+0+0=2). * 2020: Divisible by 10, but not by 3 (2+0+2+0=4). * 220: Divisible by 10, but not by 3 (2+2+0=4). * 2220: Divisible by both 10 and 3 (2+2+2+0=6).
Since 2220 is smaller than 202020 and meets all the conditions, 202020 is not the *least* such integer. The proposed answer is incorrect.

---

---

**Self-Correct Response**

1. **Divisibility by 30:** A number is divisible by 30 if it is divisible by both 2 and 3 and 5 (since 30 = 2*3*5). Since divisibility by 2 and 5 is guaranteed by divisibility by 10 (which means the number ends in 0), we need a number ending in 0 where the sum of its digits is divisible by 3.
2. **Building the number:** We want the smallest positive integer, so we start constructing the number with the fewest digits possible.
* **One 2:** 20. Sum of digits is 2, not divisible by 3. * **Two 2s:** 220. Sum of digits is 4, not divisible by 3. * **Three 2s:** 2220. Sum of digits is 6, which is divisible by 3.
3. **Verification:** 2220 is divisible by 10 (ends in 0) and by 3 (sum of digits is 6). Therefore, it is divisible by 30. It only uses the digits 0 and 2.
$\boxed{2220}$

---

## F SETS vs. Combining Sequential/Parallel

Table 6 compares the proposed SETS with the Combining Sequential/Parallel approach from Snell et al. (2024), highlighting their key differences.

| Dimension | Combining Sequential/Parallel | SETS |
|---|---|---|
| Verification | Train a task-specific reward model as the verifier. | Use the LLM itself to verify the correctness of the answer. |
| Revision | Train a revision model to predict new answers using previous answers as context. At test time, sample a sequence of revisions from the model with four previous answers as context. | Use the LLM itself to generate a new answer based on previous answers and their self-verification analyses. |
| Early Stopping in Revision | Produce a fixed number of new answers without early stopping. | The self-correction process will stop when the answer is verified as a correct one. |
| Final answer selection | Use the verifier to select the best answer within each chain and across chains. | Apply majority voting on the new responses produced by each branch of Self-Verify and Self-Correct iterations to get the final answer. |
| Evaluation | Only evaluated on MATH with PaLM 2-S. | Evaluated on six challenging benchmarks spanning planning, reasoning, math, and coding with both non-thinking and thinking models to demonstrate the generalization. |
| Practicality | Hard to be used in practice since we need to collect data to train the verifier and revision model for each task. | Can be easily applied to different tasks and unlock a lot of downstream applications. |

Table 6: Comparison of the proposed SETS with the Combining Sequential/Parallel approach (Snell et al., 2024).

