# OpenReview forum: "SETS: Leveraging Self-Verification and Self-Correction for Improved Test-Time Scaling"
_TMLR — Accepted by TMLR_

### Review · Reviewer_v4g2 · 2025-09-12

**Summary Of Contributions:**

Summary:

This paper proposes Self-Enhanced Test-Time Scaling (SETS), a test-time scaling method for better and more accurate outputs. This method combines major voting, self-verify, self-correct and shows promising performance against several test-time scaling baseline on strong close-source LLMs.

Strength:

1. The question this paper investigate is very important that how to scale better sampling via improved test-time computations.

2. The presentation is clean and clear, which make it easy to read and follow.

3. The ablation study is complete and detailed. The authors include four baselines, while SETS outperform against them when the number of tokens scale to some extent. The ablation is corresponding to the claims and statements.

4. The method SETS show better test-time scaling potential against the baseline, as shown in the Figure 2, 3.

Weakness:

1. The novelty of SETS is somehow limited. It seems a combination of the baselines (Self-Verify, Self-Correct, ...).

2. This paper mainly focuses on strong close-source models like GEMINI, but it remains a question whether SETS could be equally effective on the weaker open-source models, as they are cheaper and the initial points are weaker than the close-source. If SETS could scaling up more on these worse starting points (such as LLaMA-3 series, Qwen3 series) and even reach the level of close-source models, it will be very interesting.

3. This paper lacks an experiments between SETS and thinking mode. As thinking models are proven to be strong reasoners nowadays[1][2], whether SETS could perform better than this strong and verified baseline - thinking mode under the same token consumption remains a question. I understand it may be hard to outperform thinking mode, but this comparison could make the ablation more complete.

Questions:

1. This paper is mainly based on strong close source model, which has a good initial point. What if begin from a sub-optimal open-source model, such as LLaMA-3 series or Qwen3 series?

2. In Figure 2, when the average number of output tokens are the same, is the number of N in BoN different?

3. As this work focuses on how to obtain better and more accurate responses by scaling test-time computation, I suggest the author to conduct a comparison experiment between the non-thinking mode with SETS and the thinking mode under the same token numbers (for example, I know both Qwen3 and GEMINI-2.5-Flash has thinking mode and non-thinking mode).

[1] Guo D, Yang D, Zhang H, et al. Deepseek-r1: Incentivizing reasoning capability in llms via reinforcement learning[J]. arXiv preprint arXiv:2501.12948, 2025.

[2] Yang, An, et al. "Qwen3 technical report." arXiv preprint arXiv:2505.09388 (2025).

**Audience:**

Yes

**Audience Explanation:**

This paper is very interesting and completed.

**Claims And Evidence:**

Yes

**Claims Explanation:**

See the strength, weakness and questions.

**Requested Changes:**

Please see the weakness and questions.

---

> ### Author Response · Authors · 2025-10-09
>
> - The novelty of SETS is somehow limited. It seems a combination of the baselines (Self-Verify, Self-Correct, ...).
>
> Thank you for this insightful comment. While SETS integrates concepts like self-verification and self-correction, its novelty lies in two key areas.
>
> First, SETS introduces a unified, self-contained framework that uses the same base LLM for all stages – sampling, verification, and correction. This is a critical distinction from prior works like [1] that require separate, specially trained models for verification or revision. By demonstrating that a single model's intrinsic capabilities are sufficient, SETS offers a more accessible and practical approach, removing the significant overhead of training and deploying auxiliary models.
>
> Second, our work provides the first rigorous, large-scale evaluation of this specific integrated strategy. We systematically demonstrate how and why this simplified combination achieves a superior trade-off between accuracy and computational cost across a wide array of tasks and models. This analysis clarifies the most efficient way to leverage a single LLM's full potential at test time, providing a valuable blueprint for real-world applications.
>
> [1] Snell, Charlie, et al. "Scaling llm test-time compute optimally can be more effective than scaling model parameters." arXiv preprint arXiv:2408.03314 (2024).
>
> ----
>
> - It remains a question whether SETS could be equally effective on the weaker open-source models (such as LLaMA-3 series, Qwen3 series), as they are cheaper and the initial points are weaker than the close-source.
>
> To address the reviewer's question on open-source models, we conducted new experiments using Qwen3-235B-A22B across four diverse benchmarks (planning, reasoning, math, and coding). The results, detailed in Appendix D.2 (Figure 10), confirm that SETS consistently and significantly outperforms all baselines. This demonstrates that our method remains highly effective for weaker open-source models, successfully improving their performance even when starting from less accurate initial outputs. These findings validate that SETS is a robust technique for enhancing both state-of-the-art proprietary models and more accessible open-source alternatives.
>
> ----
>
> - This paper lacks an experiment between SETS and thinking mode: conduct a comparison experiment between the non-thinking mode with SETS and the thinking mode under the same token numbers.
>
> We thank the reviewer for this excellent suggestion. We conducted the requested experiment comparing the GEMINI-2.5-Flash-Lite non-thinking mode with SETS against the thinking mode with BoN+Majority Vote under a fixed token budget. The results, presented in Table 5 (Appendix D.10), show that the non-thinking mode with SETS does not outperform the thinking mode baseline. This is an important finding, as it clarifies that test-time scaling methods like SETS are designed to amplify a model's existing capabilities, not to bridge a fundamental gap in reasoning power between different base models.
>
> The true value of our method is demonstrated when it is applied to the most capable model available. As demonstrated in Table 5 (Appendix D.10), applying SETS to the thinking mode itself yields substantial performance gains over the thinking mode with BoN+Majority Vote. This confirms that SETS is a powerful and flexible tool for maximizing the performance of any given model. Instead of closing the gap between different models, its practical utility lies in pushing the performance ceiling of a chosen model – including the state-of-the-art thinking mode – even higher.
>
> ----
>
> - In Figure 2, when the average number of output tokens are the same, is the number of N in BoN different?
>
> The number of N in Best-of-N (BoN) might be different when the average number of output tokens are the same in Figure 2. For a given point on the x-axis (average number of output tokens), the value of N in BoN is different for each aggregation method.
> Here’s a breakdown of why:
>
> The output token cost, which is plotted on the x-axis in Figure 2, is the sum of two components:
>
> **Generation Cost**: The output tokens produced when generating the initial N solutions.
>
> **Aggregation Cost**: The output tokens produced by the method used to select the final answer.
>
> The aggregation cost varies significantly between the methods:
>
> **BoN + Majority Vote**: The aggregation step (voting) is done offline and has a output token cost of zero. Therefore, its total cost is purely the generation cost.
>
> **BoN + Self-Verify / Self-Eval**: The aggregation step involves additional LLM calls that produce its own output tokens. This step has a non-zero output token cost.
>
> To keep the total output token cost the same across all methods, the number of initial generations (N) must be adjusted. For Self-Verify and Self-Eval, N must be smaller to budget for the output tokens used in their aggregation steps.

---

### Review · Reviewer_vRha · 2025-09-19

**Summary Of Contributions:**

This paper motivates that existing test-time scaling strategies for enhancing the performance of LLMs on complex reasoning tasks have key limitations: parallel methods like repeated sampling are inefficient and saturate quickly, while sequential methods like SELF-REFINE struggle to improve after a few iterations. To address these issues, this paper proposes to combine parallel and sequential techniques. The proposed approach first generates multiple candidate solutions in parallel. Then, for each candidate, it iteratively applies the LLM's own self-verification and self-correction capabilities to refine the solution. The final answer is selected via majority voting from the set of improved candidates. The paper presents comprehensive experimental results on a diverse set of benchmarks, demonstrating the proposed approach is more effective compared to several baselines.

**Strengths**

- The problem of effective test-time scaling for LLMs is both trendy and important.
- The proposed approach is sensible, clearly described, and logically combines the strengths of parallel and sequential methods.
- The empirical evaluation is extensive and convincing in my opinion.


Note: As I am not deeply familiar with the literature on test-time scaling of LLMs, I cannot fully assess the novelty of the proposed approach relative to all prior work nor the significance of the empirical results.

**Audience:**

Yes

**Audience Explanation:**

See strengths 1.

**Broader Impact Concerns:**

No concerns.

**Claims And Evidence:**

Yes

**Claims Explanation:**

See strengths 2 and 3.

**Requested Changes:**

**Suggestions**

- Deeper Analysis of Self-Verification Reliability. The proposed approach’s effectiveness relies heavily on the LLM's ability to self-verify its answers correctly. It would be beneficial to investigate the failure modes of this core mechanism. A qualitative analysis of cases where self-verification produces false positives or false negatives would offer valuable insight into the method's limitations and guide future improvements in making the self-critique process more robust.
- Justification for Prompt Design Choices. The prompts used to perform self-verification and self-correction are critical to the method's implementation. While the prompts are provided in the appendix, the paper could be strengthened by including a brief discussion of the design philosophy behind them. Justifying why certain structural choices were made  would enhance reproducibility and clarity of the performance gains.

---

> ### Author Response · Authors · 2025-10-09
>
> - Deeper Analysis of Self-Verification Reliability. The proposed approach’s effectiveness relies heavily on the LLM's ability to self-verify its answers correctly. It would be beneficial to investigate the failure modes of this core mechanism. A qualitative analysis of cases where self-verification produces false positives or false negatives would offer valuable insight into the method's limitations and guide future improvements in making the self-critique process more robust.
>
> To better understand the limitations of our method, we conducted a qualitative analysis of the self-verification mechanism's primary failure modes: false positives and false negatives. Our review, supported by examples in Appendix D.12, indicates that these errors predominantly stem from model hallucinations, where the model generates factually incorrect or nonsensical reasoning. This analysis suggests that the reliability of self-verification is closely tied to the model's underlying factual grounding and logical consistency.
>
> ----
>
> - Justification for Prompt Design Choices. The prompts used to perform self-verification and self-correction are critical to the method's implementation. While the prompts are provided in the appendix, the paper could be strengthened by including a brief discussion of the design philosophy behind them. Justifying why certain structural choices were made would enhance reproducibility and clarity of the performance gains.
>
> Our design philosophy for the prompts centered on minimalism and generalizability to demonstrate that our method's performance is robust and not dependent on extensive prompt engineering. We intentionally created simple, standardized templates to show that the core logic of SETS is effective across diverse tasks and models without highly tailored instructions. For example, the Self-Correct prompt uses a direct instruction to “outline your step-by-step thought process for deriving a new solution.” For Self-Verify, we found a simple, structured format – asking the model to first “1. List all constraints in the TASK” and then “2. Verify if the PROPOSED SOLUTION satisfies each of the constraints” – was consistently effective at guiding the model’s reasoning. This minimalist approach enhances the reproducibility of our method and confirms that its gains stem from its inherent structure rather than from fine-tuned prompts. These points are now clarified in Appendix B.

---

### Review · Reviewer_n2dZ · 2025-09-28

**Summary Of Contributions:**

This paper proposes Self-Enhanced Test-Time Scaling (SETS), a framework that integrates self-verification and self-correction to improve inference-time scaling of LLMs. Unlike prior approaches that rely either on repeated sampling or iterative refinement, SETS combines both without requiring additional reward or revision models. The proposed method is simple yet effective. Experiments on diverse benchmarks using multiple LLMs (Gemini-1.5, Gemini-2.5, Claude-3.5) demonstrate consistent improvements in test-time efficiency and scaling laws. The method is conceptually simple, training-free, and achieves promising improvement over strong baselines.

**Audience:**

Yes

**Audience Explanation:**

Both LLMs and test-time scaling are highly popular topics with broad appeal across the research community.

**Claims And Evidence:**

Yes

**Claims Explanation:**

The motivation is technically sound. The proposed method is evaluated as effective under strong Gemini-series models, but more results on other open-source LLMs are also preferred.

**Requested Changes:**

More Results or Discussions:

The results are mainly conducted on Gemini-series models, could the authors also investigate the effectiveness on more open-resourced models like Qwen, Llama, and DeepSeek-series models? This could help further strengthen the effectiveness of the proposed method.

I am also curious about the performance of LLMs with weaker performance (like 1.5 B models). This is not a limitation, but more discussions would be much better.

While token-level cost comparisons are provided, the practical runtime and financial cost (especially for commercial APIs) discussions are preferred.

Would prompt design affect the performance？

Could other aggregation methods beyond majority voting, like self-consistency variants, confidence-weighted voting,  further boost performance?

---

> ### Author Response · Authors · 2025-10-09
>
> - Investigate the effectiveness of SETS on more open-resourced models like Qwen, Llama, and DeepSeek-series models?
>
> We conducted additional experiments using Qwen3-235B-A22B on four benchmarks, covering planning (Trip Planning), reasoning (LiveBench Reasoning), math (MATH 500), and coding (LiveCodeBench TestOutputPred). The results demonstrate that SETS consistently outperforms the baselines across all these tasks. We have included these results in Appendix D.2 (Figure 10).
>
> ----
>
> - Evaluate SETS with weaker LLMs (like 1.5B models) and provide more discussions about this.
>
> We appreciate the suggestion to explore the limits of SETS on smaller models. In response, we conducted new experiments using Qwen2.5-1.5B-Instruct across four diverse benchmarks. The results, now in Appendix D.2 (Figure 11), present a nuanced picture. SETS provides a clear benefit over baselines on tasks where the model can generate plausible solutions and critically evaluate them, such as LiveBench Reasoning and LiveCodeBench TestOutputPred. However, its performance varies on more complex tasks: it underperforms BoN+Majority Vote on MATH 500, and on the highly difficult Trip Planning benchmark, all methods failed entirely, as the base model lacked the core capacity to generate even one valid solution.
>
> This mixed performance offers valuable insight into the operational boundaries for self-improvement techniques on smaller models. Our analysis suggests that for SETS to be effective, the base model must possess a foundational, even if imperfect, ability to both generate plausible solutions and critically evaluate them. On tasks where the 1.5B model has this latent competence (reasoning and coding), SETS successfully amplifies it. However, when the model's self-verification capability is unreliable (as is likely on complex math problems) or the task is fundamentally beyond its knowledge threshold (as with Trip Planning), the benefits diminish. This clarifies that while SETS is a powerful amplifier, it is not a substitute for a model's core knowledge.
>
> ----
>
> - While token-level cost comparisons are provided, the practical runtime and financial cost (especially for commercial APIs) discussions are preferred.
>
> Thank you for this valuable feedback. We agree that a discussion of practical costs is essential and have clarified this in the paper.
>
> **Financial Cost**: We directly address the practical financial cost in Appendix D.9, where we plot model accuracy against the cost in US dollars based on commercial API pricing. As shown in the test-time scaling curves, our method SETS offers a significantly better accuracy-to-cost trade-off, demonstrating its superior financial efficiency in a real-world scenario. We have made this point more prominent in the main manuscript.
>
> **Runtime Analysis**: Regarding practical runtime, we opted for a more stable and reproducible metric. Wall-clock time can be highly volatile and difficult to compare fairly, as it's heavily influenced by transient factors like network latency, server load, and specific hardware. To provide a standardized and hardware-agnostic comparison, we use API token counts and financial cost as a direct proxy for computational expense. This approach ensures our results reflect the intrinsic efficiency of the methods themselves, rather than the particulars of one experimental environment. We have clarified this in the revised manuscript.
>
> ----
>
> - Would prompt design affect the performance？
>
> Thank you for this important question. Yes, prompt design is known to affect LLM performance, and we took a deliberate approach to it in our study.
>
> Our primary goal was to demonstrate that our method, SETS, is robust and generalizable across diverse tasks and models. To this end, we intentionally used minimal and standardized prompt templates rather than pursuing extensive, task-specific prompt engineering. This approach allows us to fairly evaluate the core capabilities of our method, rather than the effects of highly optimized prompts.
>
> We found that simple, structured instructions were broadly effective: For Self-Correct, we used a straightforward prompt asking the LLM to “outline your step-by-step thought process for deriving a new solution.” For Self-Verify, we discovered that a simple two-step instruction consistently improved performance: “1. List all constraints in the TASK. 2. Verify if the PROPOSED SOLUTION satisfies each of the constraints.”
>
> This minimalist strategy demonstrates that SETS does not rely on fragile, heavily engineered prompts to work effectively. While we acknowledge that further gains could likely be achieved by tailoring prompts to each specific task, our work establishes a strong, generalizable baseline. The templates for these prompts are provided in Appendix B for reproducibility.

---

> ### Author Response · Authors · 2025-10-09
>
> - Could other aggregation methods beyond majority voting, like self-consistency variants, confidence-weighted voting, further boost performance?
>
> That is an excellent suggestion. To investigate the potential of more sophisticated aggregation strategies, we conducted new experiments evaluating SETS with confidence-weighted voting. This approach moves beyond a simple majority vote by weighting each candidate solution with a confidence score, which we define as the proportion of times the solution is verified as correct during the scaling process.
>
> The results in Appendix D.11 show that this method generally enhances performance over standard majority voting, achieving superior results on most benchmarks (Trip Planning, MATH 500, LiveBench Reasoning, and LiveCodeBench TestOutputPred). However, this improvement is not universal. On certain tasks, such as Meeting Planning and AIME 2024-2025, the simpler majority vote remains more effective, suggesting that the reliability of self-verification scores as a confidence heuristic can be task-dependent. These results also show that SETS can be integrated with diverse aggregation methods, demonstrating its generalizability.

---

### Author Response · Authors · 2025-10-09

We thank the reviewers for their constructive feedback and have thoroughly revised the manuscript to address their concerns. All major revisions are highlighted in blue. The key changes include:

- Evaluated SETS on an open-source model (Qwen3-235B-A22B) to demonstrate its generalizability beyond proprietary APIs (Appendix D.2, Figure 10).
- Investigated the performance of SETS on a small 1.5B model (Qwen2.5-1.5B-Instruct) to analyze its effectiveness and limitations on weaker LLMs (Appendix D.2, Figure 11).
- Added a practical financial cost analysis using API pricing in dollars, directly addressing concerns about real-world deployment costs (Appendix D.9).
- Clarified our rationale for using token counts over wall-clock time as a more stable and reproducible metric for computational expense (Section 4.2).
- Explored alternative aggregation strategies by adding experiments with confidence-weighted voting, as suggested by the reviewers (Appendix D.11).
- Provided a deeper qualitative analysis of self-verification's failure modes (false positives/negatives) to better characterize the method's limitations (Appendix D.12).
- Added a detailed justification for our prompt design philosophy, explaining the rationale behind our minimalist and structured approach (Appendix B).
- Conducted a new experiment to clarify the role of SETS, showing it amplifies a given model's capability (e.g., "thinking mode") rather than closing the gap between different models (Appendix D.10).

---

### Decision · Action_Editor_53fP · 2025-11-08

**Recommendation:** Accept with minor revision

**Additional Comments:**

The paper is sound and well-evaluated. Minor revisions could include clarifying the novelty gap vs. prior work like SELF-REFINE and tightening the discussion on weaker open-source models.

**Audience:**

Yes

**Audience Explanation:**

Test-time scaling is a highly relevant topic for TMLR’s audience. The paper proposes a training-free method (SETS) that enhances LLM performance via self-verification and self-correction, which is of broad interest to researchers working on inference-time efficiency and LLM reasoning.

**Claims And Evidence:**

Yes

**Claims Explanation:**

The authors present extensive empirical results across diverse benchmarks (planning, reasoning, math, coding) and multiple models (Gemini, Claude, Qwen). Ablation studies, cost analyses, and comparisons with baselines like BoN and SELF-REFINE are included. Reviewers confirmed the evidence is convincing, though some noted weaker gains on non-Gemini models.

---

> ### Author Response · Authors · 2025-11-24
>
> Dear Action Editor,
>
> Thank you for the acceptance and the constructive feedback. We have now submitted the camera-ready version of Paper 5774.
>
> In this revision, we have addressed your specific requests by:
> - Clarifying the novelty gap between SETS and prior work like SELF-REFINE (Section 3).
> - Tightening and expanding the discussion regarding performance on weaker open-source models (Appendix D.2).
>
> We hope these updates satisfy the requirements. Please let us know if any further adjustments are needed.
>
> Best regards,
>
> Paper 5774 Authors